# Identification and Clarification of *VrCYCA1*: A Key Genic Male Sterility-Related Gene in Mungbean by Multi-Omics Analysis

Jinyang Liu, Yun Lin, Jinbin Chen, Chenchen Xue , Ranran Wu, Qiang Yan, Xin Chen and Xingxing Yuan *

Institute of Industrial Crops, Jiangsu Academy of Agricultural Sciences/Jiangsu Key Laboratory for Horticultural Crop Genetic Improvement, Nanjing 210014, China; 20200024@jaas.ac.cn (J.L.); 18180006@jaas.ac.cn (Y.L.); chenjingbin@jaas.ac.cn (J.C.); xuecc@jaas.ac.cn (C.X.); rrwu@jaas.ac.cn (R.W.); yanqiang@jaas.ac.cn (Q.Y.); cx@jaas.ac.cn (X.C.)
* Correspondence: yxx@jaas.ac.cn (X.Y.)

**Abstract:** Heterosis has been utilized in crops for a long time, and although crop fertility is the basis for the utilization of heterosis, there is limited information concerning the genic male sterility (GMS) of mungbean. Therefore, based on the genic male sterile mutant, $M_1$, obtained by EMS mutagenesis of the Weilyu cultivar, this project used multi-omics analysis to detect the male sterile site and further identify its candidate gene, and then the mechanism of the candidate gene was discussed. As a result, one locus region (Chr5: 6,835,001–6,935,000 bp) associated with GMS was identified, and nine genes were found within the 100 Kb region. The candidate gene, *VrCYCA1*, around the above loci had a TATA box deletion approximately 4.7 Kb upstream of the gene, and this was evidenced by comparative genomics, transcriptome analysis, and RT-qPCR analysis. The expression level of *VrCYCA1* was significantly downregulated ($\log_2$FC = −2.06, *p*-value = 0.025) in the *ms* lines compared with the control group. Moreover, 6653 genes showed differential expression between the Weilyu lines and mutant lines as well as 165 metabolites with significant differences in their concentration levels. Among those differentially expresses genes, 226 were annotated with functional categories involved in flowering and endosperm development, and six genes had protein–protein interactions with *VrCYCA1*. Seven categories of metabolites and seven genes participated in the relationship between reproductive growth and vegetative growth, which might have caused the sterility of mungbean in the mutant plants. This study used multi-omics data to mine a mungbean GMS-related gene, *VrCYCA1*, and constructed a GMS genetic network to explore the molecular mechanism of *VrCYCA1*. The results lay a solid foundation for further molecular biology research and utilization in mungbean male sterility.

**Keywords:** GMS; multi-omics analysis; BSA; transcriptome; metabonomics; RT-qPCR; *VrCYCA1*



## 1. Introduction

Genic male sterility (GMS) is an important tool to study crop heterosis, as well as quantitative traits, generally controlled by nuclear genes [1]. The phenomenon of plant GMS widely exists in nature, and the genic male sterile dual-purpose line bred with it can be used as both a male sterile line and a maintainer line, which makes it a prominent line of work in crop heterosis utilization. Mining plant GMS-related genes has become important work in crop heterosis utilization.

To date, many genes have been reported to be involved in male sterility in *Arabidopsis*, rice and maize. At present, more than 100 GMS-related genes have been reported [2,3]. In *Arabidopsis*, more than 30 GMS-related genes have been cloned such as *MMD1*, *ROXY1/ROXY2*, *MYB80*, *NEF1*, *CYCA1*, *rpg1*, *AtSec62* and *ms157* [4–8]. In rice, more than 30 GMS-related genes have been reported such as *OsFIGNL1*, *OsFTIP7*, *OsUAM3*, *OsGT1*, *OsAPI5*, *OsTGA10* and *RMS2* [9–13]. In maize, there are relatively few GMS-related genes that have been cloned such as *MAC1*, *ms9*, *Ocl4*, *Ms7*, *ms26*, *ms30*, *ipe2*

and *bHLH51* [14–19]. These genes are mainly involved in plant meiosis, pollen development, tapetal cell development, cuticle membrane and wax production, lipid metabolism, and polysaccharide metabolism progress [20]. Some genes participate in the DNA methylation system, which is required for proper pollen tube orientation [21]. However, there are few reports on mungbean (*Vigna radiata* (L.) Wilczek) GMS-related genes and much fewer regarding genes and metabolites involved in the GMS process that directly affects the process of mungbean cross-breeding. Therefore, identification of GMS-related genes has practical significance for the genetic improvement of mungbean.

In the genetic analysis of quantitative traits, the most commonly used methods are genome-wide association study (GWAS), linkage analysis and bulk segregation analysis (BSA). Among these methods, linkage disequilibrium (LD) is the theoretical basis of association analysis [22–24]. GWAS is generally applicable to the study of the quantitative trait nucleotides/loci (QTNs/QTL) in natural populations, and linkage analysis is widely used in $F_2$ ($F_2$ population), BC (backcross population), DH (doubled-haploid population), RIL (recombinant inbred line) and NAM (nested association mapping) population analysis, and BSA is mainly used in $F_2$ population and EMS (ethyl methane sulfonate) mutagenesis material [25]. To date, thousands of QTNs/QTL involved in quantitative traits have been identified by GWAS and linkage analysis.

BSA is a highly efficient and rapid QTL mapping method [26], especially for the mapping of single-gene qualitative characters or major-gene quantitative traits [25]. As mutants and wild-type individuals are both homozygous, BSA also has obvious advantages in the mapping of EMS-induced mutant genes, and it helps to reduce the time and cost. In the QTL mapping of the segregation population, for example, Takagi et al. (2013) [27] identified a candidate gene, *osrr22*, controlling salt tolerance in rice using whole-genome resequencing and MutMap analysis. Chai et al. (2021) [28] identified *BnA10g0422620* and *BnA10g0422610* as candidate genes related to the blade-like outgrowth trait in rapeseed. Feng et al. (2021) [29] identified the restorer gene, *Rf2*, in cytoplasmic male sterile cotton. For EMS mutagenesis material, this method also showed applicability in the localization of candidate genes; for example, Fekih et al. (2013) [30] used the method to identify *Os08g0139100* as a candidate gene responsible for albinism and early death. Chen et al. (2014) [1] cloned *MER3*, a rice male sterility gene, using the MutMap method (a method based on whole-genome resequencing of pooled DNA from a segregated population of plants that show a useful phenotype).

Usually, linkage analysis trends map a significant locus in a large chromosome interval, and there is often more than one gene in the interval. During candidate gene identification, multi-omics integrative analyses are usually used to find the main genes. In this study, 284 and 279 candidate genes were found by phenotypic and metabolic genome-wide association studies and multi-omics analyses, respectively, to be significantly associated with seed-oil-related traits and metabolites in soybean [31]. All the genes around the QTL were used to mine their *Arabidopsis* homologous genes that were annotated in target trait-related pathways using comparative genomics analysis. If there are more than 10 genes in the interval, transcriptome and RT-qPCR analysis can be used in the candidate gene identification [32,33]. Metabolites are the basis of agronomic traits and act as a bridge between traits and genes [34]. In complex trait genetic network analyses, it is necessary to comprehensively analyze the possible relationships between gene, metabolite and traits [35–37], Especially in the transition from vegetative to reproductive growth, metabolites play a crucial role, e.g., lipids and flavonoids [38,39]. At present, studies on mungbean GMS are relatively limited, and mining GMS-related genes and analyzing their related genetic networks will provide an important practical basis for genetic improvement and heterosis utilization of mungbean.

To address the above issues, two BSA data sets were used to identify the significant QTNs associated with GMS. One consisted of male sterile plants and fertile plants of $M_3$ and another consisted of male sterile plants and fertile plants of $M_4$, and the male sterile sites were further identified by the SNP-index and Euclidean distance (ED) methods. Stable

male sterile sites were the common QTNs, obtained by combining the two results above, and candidate genes of male sterility were identified by combining comparative genomics, transcriptome and RT-qPCR analyses. Moreover, the GMS-related genetic networks were established by identification of differentially expressed genes, differential metabolites and PPI (protein–protein interaction) analyses. According to the results obtained, a *VrCYCA1*-related GMS genetic network was constructed, which could reveal mungbean GMS of the male sterile mungbean (*ms*), from the perspective of genes and metabolites. The key candidate genes identified in this study would be useful for mungbean quality improvement and gene function identification.

## 2. Materials and Methods

### 2.1. Plant Material

The *ms* lines were obtained by EMS (40 μL/mL for 2 h with gentle agitation) mutagenesis of Weilyu 11 (cultivar) seeds. All of the Weilyu 11 (CK) and *ms* lines were planted in an experimental field at the Institute of Industrial Crops, Jiangsu Academy of Agricultural Sciences, in 2020 and 2021. Each plot had three rows, and a single row was 2.5 m long with a 15.0 cm intra-row spacing and a 60 cm inter-row spacing. The plots were 1.2 m wide and 2.5 m long.

### 2.2. Genetic Analysis of Mungbean Male Sterile Mutants

The fertile plants from the same plant as the sterile plants were harvested per plant, and the heterozygous fertile plants in $M_2$ were used to breed $M_3$ and $M_4$. After removing the phenotypic with the nonsegregated individuals in $M_2$, the segregated population in $M_2$ then formed a genetic population that could be used for genetic analysis. Then, $M_3$ and $M_4$ were bred from a single segregation of heterozygous fertile plants in $M_2$. The number of fertile and sterile individuals in the segregated population were counted (Table 1), and the conformity was statistically analyzed using the chi-square test with $p < 0.05$ set as being statistically significant, where $2 < 3.841$. The lengths of the flower stalks of Weilyu 11 and *ms* were counted in the $M_4$ generations ($n = 5$), and Student's *t*-test statistical analyses were utilized to determine differences in flower stalk lengths, with $p < 0.05$ set as being statistically significant [40].

**Table 1.** Genetic analysis of the fertility separation population in the $M_4$ generation.

| Phenotype | Observed Number | Theoretical Number | $\chi^2_{(3:1)}$ |
|---|---|---|---|
| Fertile plants | 177 | 165 | 3.491 |
| Sterile plants | 43 | 55 | |

$\chi^2 = 3.841$; $p = 0.05$.

### 2.3. Viability Test of Mature Pollen Grains and Phenotypic Investigation

The flower buds and blooming flowers of the CK and *ms* mutants were collected at different development stages, during the full flowering stage of mungbean at approximately 8:00–9:00 a.m. The fixed mature pollen was separated with tweezers and placed on a slide with 1–2 drops of 1% $I_2$-KI solution added, and then the slide was covered and observed directly under an optical microscope [13,17]. For each line, three pollen samples of CK and *ms* were taken for observation. More than 200 pollens grains were counted, and the abortion rate of pollen was estimated by observation of their staining rate. Photos of the flowers in the vegetative growth stage and seeds in the reproductive growth stage were used to compare the flower and seed shapes. Stalks changes between control and male sterile plants were determined by stalk length measurement.

### 2.4. Cytological Analysis

To investigate the pollen of the *ms* lines, the flowers at 0 DAF (days after flowering) were examined by serial paraffin sections. Three flowers from each line of *ms* and CK were

sampled. The flowers were first soaked in distilled water for 20 h and then fixed in 50% FAA solution using 70%, 90% and 100% ethanol to dehydrate the FAA solution (formalin 5 mL + acetic acid 5 mL + 70% ethyl alcohol 90 mL) for 15 min. Semi-thin (2.0 μm) sections were obtained using an automatic microtome, stained with 0.1% toluidine blue O for 30–60 s at room temperature. The detailed experimental procedures have been described in previous studies [41].

To investigate the detailed morphology of pollens, the flowers at 0 DAF were examined by scanning electron microscope analyses. For each line, the flowers were first put in the FAA fixative for 2 h, then transferred to a refrigerator at 4 °C and washed in 0.1 M PBS (phosphate buffer saline) (pH 7.4) three times for 15 min each time. Then, the flowers were post-fixed with 1% $OsO_4$ in 0.1 M PBS for 1–2 h at room temperature and kept away from light. $OsO_4$ was removed by rinsing three times in 0.1 M PBS for 15 min each time. The blocks were then washed; dehydrated through an ethanol series at 30%, 50%, 70%, 80%, 90%, 95%, 100% and 100%; dehydrated through isoamyl acetate for 15 min. The detailed experimental procedures have been described in previous studies [42]. Sections were allowed to air-dry overnight at room temperature. Samples were placed on double-sided conductive carbon film and then sprayed with gold for approximately 30 s under the ion beam sputtering instrument. The slice samples were photographed under a transmission electron micrograph (TEM, SU8100, Hitachi, Japan).

### 2.5. Whole-Genome Resequencing

The young leaves of the CK (Weilyu 11) pool (1), the male sterile plants in $M_3$ pool (2), the male sterile plants in $M_4$ pool (3) and the fertile plants in $M_4$ pool (4) (individuals > 30) were collected after viability tests of mature pollen grains for DNA extraction using the CTAB method [42]. Thus, two BSA experiments were conducted: one consisted of (1), (2) and (4); another consisted of (1), (3) and (4), and they were used for sequencing [43]. Short reads sequenced by an Illumina HiSeq 4000 platform (with an average sequencing depth $\geq 50\times$) were mapped to scaffolds using the Burrows-Wheeler Alignment Tool (BWA) (Version 0.7.15) (http://bio-bwa.sourceforge.net/bwa.shtml, accessed on 10 December 2021) [44]. The Genome Analysis Toolkit (GATK) was used to select the SNP and indel (https://gitee.com/mirrors/GATK, accessed on 15 December 2021) [45]. Sulv 1 genome was chosen as the reference genome [46] in the GATK analysis.

### 2.6. SNP-Index and ED Methods to Identify the QTNs of Male Sterility

The preprocessing procedures for the SNP-index and Euclidean distance (ED) value were as follows. (a) Only SNPs with a minor allele frequencies (MAFs) $\geq 0.05$ and a missing rate < 10% in the populations were used. (b) The allele frequency of each pool was analyzed and compared with the CK pool, and the proportion of genotypes different from the CK pool was the SNP-index of this locus. In addition, the ΔSNP-index was the difference in the SNP-index between the two mixing pools. The ΔSNP-index was calculated by a sliding window (20 Kb; the linkage disequilibrium value was at approximately 20 Kb). Moreover, the threshold for significant QTNs was set at a ΔSNP-index $\geq 0.8$ [43]. (c) For the ED value, $ED^2 = (A_1 - A_2)^2 + (T_1 - T_2)^2 + (C_1 - C_2)^2 + (G_1 - G_2)^2$. An, Tn, Cn and Gn referred to the frequency of the four bases (ATGC) in the mixed pool of n, and the threshold for significant QTNs was set at ED $\geq$ median + 3 SD [30]. In this study, there were two BSA pools: one consisted of the CK (Weilyu 11) pool (1), the male sterile plants in the $M_3$ pool (2) and the fertile plants in the $M_4$ pool (4); the other consisted of the CK (Weilyu 11) pool (1), the male sterile plants in the $M_4$ pool (3) and the fertile plants in the $M_4$ pool (4). The fertile plants in the $M_4$ pool (4) were used as the fertile plants pools for the two experiments. Significant QTNs were commonly obtained by both of the two BSA pools' analysis results, and stable QTNs were observed across multiple BSA comparisons. Significant QTNs identified both in the (1), (2) and (4) and the (1), (3) and (4) pools were considered the reliable QTNs. The Sulv 1 genome was chosen as the reference genome [46].

### 2.7. Candidate Gene Identification

Candidate genes for GMS were mined in three steps. First, all genes between the 20 Kb around the significant QTNs were mined. Second, the genes or their *Arabidopsis* homologous genes, which were annotated with cell differentiation, meiosis, pollen tube growth, pollen tube development, flavonoid biosynthesis and amino acids biosynthesis biological pathways that could cause male sterility in plants, were identified. Then, all the above genes were used to analyze the significantly expression levels in *ms* ($M_4$ pool) and CK ($M_4$ pool). Lastly, selected DEGs were verified by quantitative real-time PCR (RT-qPCR). Pollens of *ms* ($M_4$ pool) and CK ($M_4$ pool) at 0 DAF were selected for RNA-seq analysis. Total RNA was extracted using TRIzol reagent (Invitrogen, Shanghai, China) according to the manufacturer's instructions. The extracted RNA was then treated with RNase-free DNase I (Promega, Madison, WI, USA). After the reverse transcription step (QuantiTect Reverse Transcription Kit, Qiagen, Valencia, CA, USA), the cDNA was used as a template for RT-qPCR using the Takara Bio TB Green Premix Ex Taq (Tli RNase H Plus, Japan). The parameters were set as follows: step 1—95 °C for 300 s; step 2—95 °C for 20 s; step 3—55 °C for 20 s; step 4—72 °C for 20 s; step 5—go to step 2, 40 cycles; step 6—melt curve 60.0–95 °C. Reactions were run on a Bio-Rad CFX96 system. The delta-delta Ct method was used to calculate the relative expression levels [47]. Student's *t*-test statistical analyses were utilized to determine significant changes in gene expression with $p < 0.05$ set as being statistically significant. A mungbean tubulin (*EVM0007380*; homologous of *At3g18780*) gene was amplified and used as the reference gene in this experiment (Table S2), and three replicate samples were measured. Information of the primers used are presented in Table S2. Primers were designed by NCBI and tested by PCR of the tubulin.

### 2.8. RNA-Seq Analysis

Pollens of ms and CK (M4) at 0 DAF were selected for RNA-seq analysis. Total RNA was extracted using TRIzol reagent (Invitrogen, Shanghai, China) according to the manufacturer's instructions. High-quality RNA samples (OD260/280 = 1.8~2.2; OD260/230 $\geq$ 2.0; RIN $\geq$ 6.5; 28S:18S $\geq$ 1.0 and >10 μg) were used to construct the sequencing library (G9691B, Agilent). One microgram of total RNA was analyzed in an Illumina Novaseq Sequencer. The raw reads were cleaned by Trimmomatic (http://www.usadellab.org/cms/index.php?page=trimmomatic, accessed on 10 January 2022) [48], and then clean reads were mapped to reference sequences (Sulv 1 genome) using Hisat2 (https://daehwankimlab.github.io/hisat2/, accessed on 15 January 2022) [49]. The gene expression level was calculated using the reads per kilobase million (RPKM) method by featurecount [50] (http://subread.sourceforge.net/, accessed on 20 January 2022). Benjamini and Hochberg's approach was used for controlling the false discovery rate. Genes with an FDR < 0.01 and a fold change $\geq$ 2 were assigned as differentially expressed. The transcriptome raw data were uploaded to NCBI, the accession was PRJNA822679, and the ID was 822679.

### 2.9. Metabonomic Analysis

A liquid chromatography-mass spectrometry system was used to analysis the metabolites in the pollens of *ms* and CK ($M_4$) at 0 DAF. The pollens were crushed using a mixer mill (liquid nitrogen environment), and 100 mg of powder was weighted and extracted overnight at 4 °C with 1.0 mL pure methanol acetonitrile water (1:1). One microliter of the prepared sample was injected into the LC-20AD system (HPLC, Shimadzu LC-20AD, Japan). The detailed instrument operating parameters have been described in previous studies [31]. The Madison Metabolomics Consortium Database (MMCD) was used to identify metabolites, the m/z values of each metabolites were blast with MMCD [51]. Three pollen samples of CK and *ms* were taken for detection. Student's *t*-test statistical analyses were utilized to look for significant changes in metabolites content with $p < 0.05$ set as being statistically significance.

### 2.10. PPI (Protein-Protein Interaction)

The protein-protein interactions for candidate genes in transcriptome and BSA analyses were detected used the online tool STRING (https://string-db.org//, accessed on 30 January 2022) [52]. Significant PPIs had medium predicted values $\geq$0.40.

## 3. Results

### 3.1. Phenotypic Analysis of GMS Mutants in Mungbean

Based on the genetic male sterile mutant, $M_1$, obtained by EMS mutagenesis in Weilyu 11, there were 23 lines in total of which 6 were sterile, named "*ms*" (male sterile mungbean ms, abbreviated as *ms*). The $M_4$ generation, which was derived from $M_1$, were used to do the phenotypic analysis. There were no significant differences in plant height, plant type, flower shape and structure between *ms* and "Weilyu 11 (CK)" (Figure 1a). At maturity "Weilyu 11 (CK)" leaves turned yellow and fell off, with a large number of seed pods, while *ms* was still vigorous, the leaves were light green and did not fall off. Most importantly, the *ms* could not form normal pods (Figure 1c,d) or occasionally had small pods, but they fell off naturally 3~5 days after flowering, and the length of *ms* flower stalks (3.78 $\pm$ 0.62 cm) were significantly longer than the CK (12.73 $\pm$ 0.62 cm, *p*-value = 2.40 $\times$ 10$^{-10}$) (Figure 1e). In the detection of pollen viability, the results of $I_2$-KI staining showed that the wild-type pollen had a regular shape and could be evenly stained, while the pollen of *ms* was abnormally stained (Figure 1b) with typical abortion and round abortion. In addition, in the genetic analysis of the fertility separation population, there were 58 lines in the fertility segregation population ($M_3$ generation) of ms of which 42 were fertile and 16 were sterile. Among the $M_{3:4}$ lines derived from these 42 fertile plants, 30 lines were separated. The separation ratio of these isolated lines was counted in the $M_4$ generation. There were 177 fertile lines and 43 sterile lines, $\chi^2_{(3:1)}$ was equal to 3.491, less than 3.84 ($\chi^2_{(3:1)}$ = 3.84, *p*-value = 0.05) (Table 1). The results indicated that the male sterility trait of *ms* was controlled by a single recessive nuclear gene.

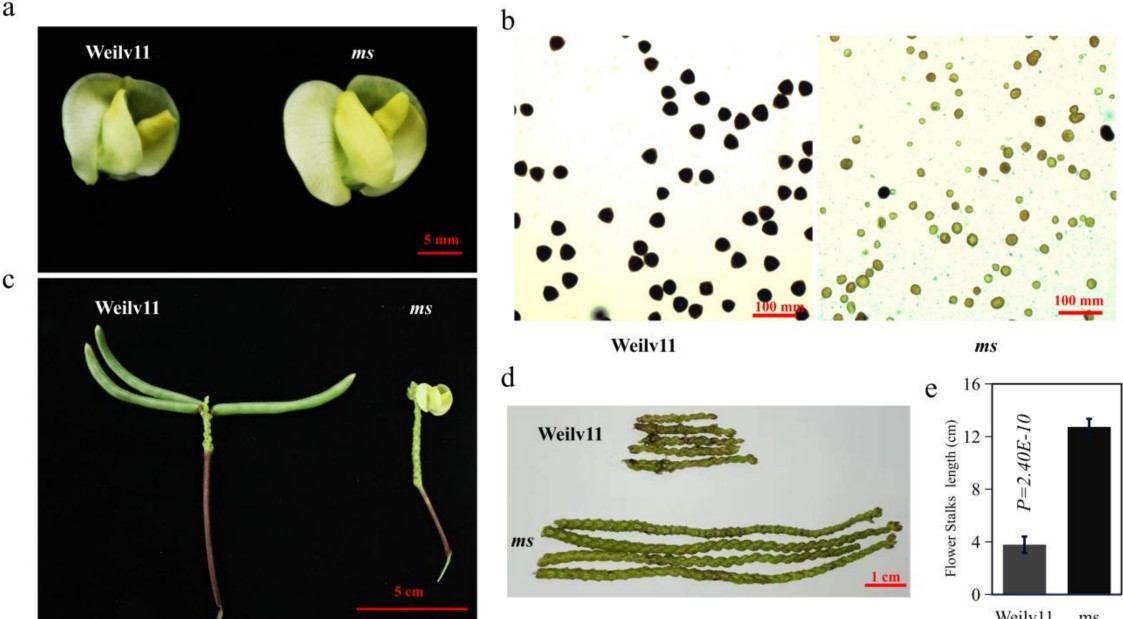

**Figure 1.** Phenotypic analysis of the *ms* mutant: (**a**) vegetative growth stage at 2 DAF; (**b**) $I_2$-KI staining of Weilyu 11 and *ms* mature pollen; (**c**) reproductive growth stage at 20 DAF; (**d**) the length of flower stalks of Weilyu 11 and *ms* at 45 DAF; (**e**) statistics on the flower stalks of Weilyu 11 and ms. In (**c**), bar = 5 mm; in (**b**), bar = 100 mm; in (**c**), bar = 5 cm; in (**d**), bar = 1 cm. Student's *t*-test statistical analyses were utilized to look for differences in the length of flower stalks with *p* < 0.05 set as statistically significance.

*3.2. Histocytological Comparison of ms with Its Wild Type*

Through the paraffin section (cross-cutting) of the flowers, it can also be seen that the wild-type pollen in the pollen sac could be evenly stained (Figure 2a,c,e), and the amount of pollen was greater than that in *ms* (Figure 2b,d,f). Moreover, the pollen in *ms* was abnormally stained, and the dyeing effects were shallow (Figure 2d,f) compared with the CK (Figure 2c,e). In our histocytological results, the anthers showed no significant difference between *ms* and CK (Figure 2g,h); however, the wild-type pollen grains were plump and nearly spherical (Figure 2i), the surface was arranged with multiple regular polygons and the germination pores were visible (Figure 2i,k). While the pollen of *ms* was irregularly round-shaped (Figure 2j), most of the pollen grains were surface-depressed (Figure 2j), and most of the germination pores were invisible (Figure 2j,l).

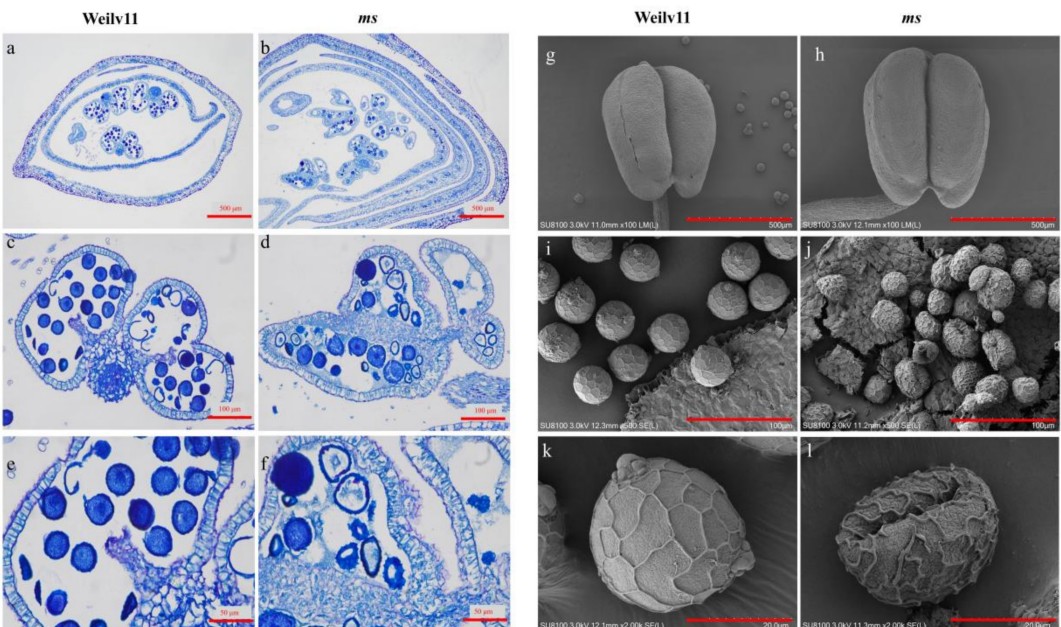

**Figure 2.** Histocytological and microscopic analyses of pollen in Weilyu 11 and ms. Longitudinal sections of other wild-type lines (background Weilyu 11 (**a**,**c**,**e**) and mutant (*ms*, (**b**,**d**,**f**)) in mature flowers under a high-resolution white camera, where the bars represent 500, 100 and 50 µm, respectively. Microscopic analyses of wild-type lines (background Weilyu 11 (**g**,**i**,**k**)) and mutant (*ms*, (**h**,**j**,**l**)) in mature flowers, where the bars represent 500, 100 and 20 µm, respectively. Both (**c**,**e**) and (**d**,**f**) are enlarged images of (**a**,**b**); (**k**,**i**) sections of the images in (**g**,**h**), respectively. (**a**–**f**) Glowers sampled at 0 DAF.

*3.3. Detection of Locus and Gene for GMS in Mungbean*

In order to detect the male sterile site and further identify its candidate genes, two BSA pools consisting of the male sterile plants and fertile plants of $M_3$ and $M_4$ were used to form the sterile pool and fertile pool, respectively. In total, 196,869,736; 213,597,502; 193,903,636; 200,865,064 clean reads were obtained from the CK (Weilyu 11) pool, the male sterile plants in the $M_3$ pool, the male sterile plants in the M4 pool and the fertile plants in the $M_4$ pool, respectively, with an average sequencing depth $\geq 50\times$ (Table S1); 500,315 and 53,529 SNPs (Supplementary Materials Data S1 and S2) were detected in the $M_3$ and $M_4$ BSA data, respectively, compared to the Sulv 1 genome, which was chosen as the reference genome.

In the SNP-index and ED analysis, a stable QTL was detected on Chr5: 6,835,000~6,935,000 bp interval ($\Delta$SNP-index > 0.86, ED > 1.61) (Figure 3a) in the two BSA data sets. As was previously reported that the LD (linkage disequilibrium) value of mungbean was approximately 20 Kb, we found nine genes in the region (Figure 3b). In addition, an ATTATA box was changed into T at approximately 4.7 Kb upstream of the *VrCYCA1* gene (Figure 3c). According to the gene annotation and comparative genomics analysis, we found the function

of *EVM0001741* (*VrCYCA1/VrTAM*) to be a cyclin protein. CYCA1 is shown to be related to sexual reproduction of *Arabidopsis* as reported previously [53]. In this study, analysis of the RNA sequences at 0 DAF identified *VrCYCA1* to be differentially expressed between *ms* and CK flowers, and *VrCYCA1* had significantly higher expression levels in CK than in *ms* (*p* = 0.041, Figure 3d).

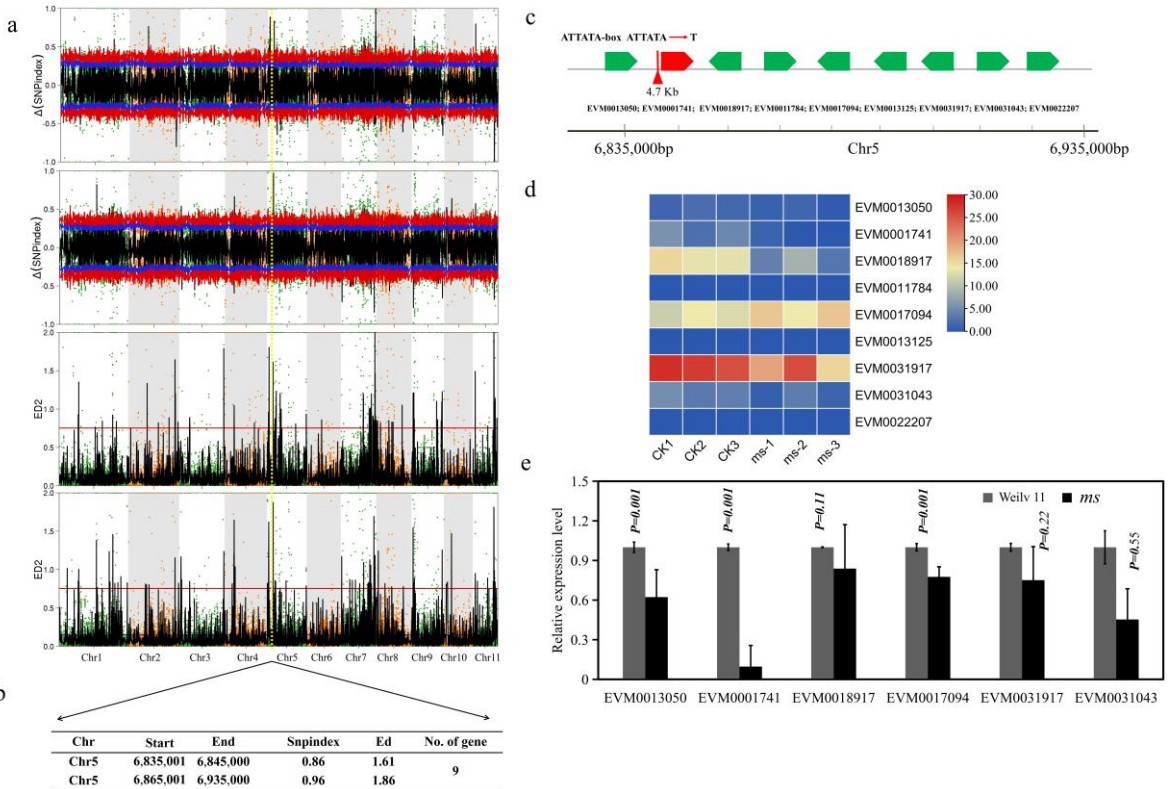

**Figure 3.** Genetic mapping of mutants and candidate genes identification: (**a**) significant QTL and (**b**,**c**) their candidate genes associated with GMS in mungbean using SNP-index and ED analyses. The critical SNP-index of significant QTL (**a**,**b**) was set at 0.8, the critical ED value of significant QTL was set at a median + 3 SD; all the candidate genes around significant QTL are listed in (**c**), and the mutation site is indicated by the red line; (**d**,**e**) expression profiling of the candidate genes for the GMS identified in this study; (**e**) real-time PCR analysis of candidate genes of mutant mungbean, and the *t*-test was used to test significant differences in gene expression between CK (Weilyu 11) and *ms*.

### 3.4. Identification of Differentially Expressed Genes and Metabolites in GMS Mutants

The expression levels of transcripts from *ms* and CK flowers were quantified using an FPKM method. A total of 6653 DEGs (fold change > 2 and a false discovery rate (FDR) < 0.01) were identified (Figure 4a,b, Table S4) of which 1779 genes were upregulated and 4874 DEGs were downregulated (Figures 3e and 4a,c, Supplementary Materials Data S4). Those genes mainly were concentrated in 30 bioprocesses including pollen tube growth, biological regulation and pollen tube development (Figure 4d and Figure S1). Two hundred and twenty-six DEGs had functions related to flower development, embryo development and seedling development including *VrCYCA1* (Supplementary Materials Data S5). We found that *VrCYCA1* had significantly higher expression levels in CK than in *ms* (*p* = 0.041, Figure 3d). Then, we conducted RT-qPCR analysis, and the results showed that among the nine genes mentioned above, *EVM0013050* (*p*-value = 0.001), *EVM0001741* (*p*-value = 0.001) and *EVM0017094* (*p* = 0.001) were down-regulated, and they were significantly differentially expressed between *ms* and CK flowers (Figure 3e, Supplementary Materials Data S3).

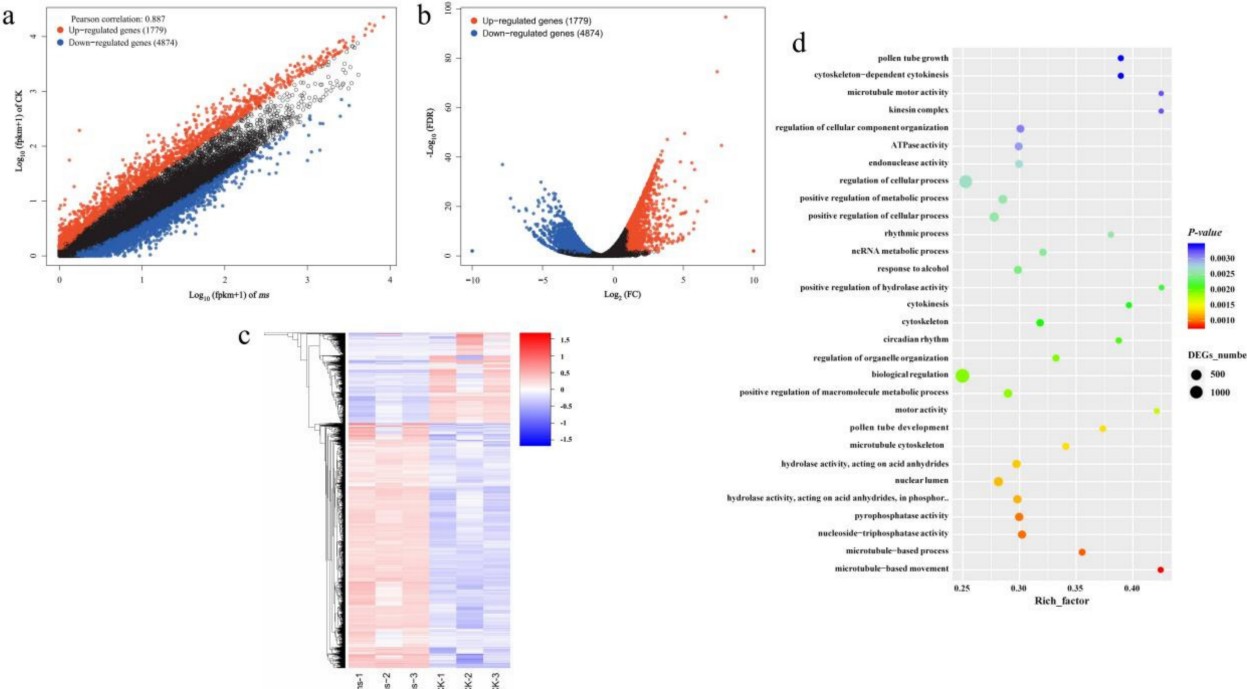

**Figure 4.** The expression profiles of the identified DEGs. Differences and characterizations in the genes' expression profiles between CK and *ms*: (**a**) scatter plots were used to evaluate the differences in the expression of genes between the CK group and the *ms* group, and the values plotted on the *x*- and *y*-axes were the averaged normalized signal values of each group (log10 fpkm + 1); (**b**) volcano plots were used for visualizing the differential expression of genes between the CK group and the *ms* group. The vertical lines correspond to a 2.0 fold (log2 fold change). Red and blue points represent significant DEGs with an FDR ≤ 0.05, log2 (fpkm + 1) > 0 (**a**) and log2 (fold change) > 1 (**b**); (**c**) the top 30 pathways of KEGG functional enrichment among DEGs; (**d**) the coloring indicates a *p*-value that is higher in blue and lower in red, and a lower *p*-value indicates a more significant enrichment. The point size indicates the number of DEGs.

The metabolite concentration in *the ms* and CK flowers were also analyzed. A total of 252 metabolites were identified, and 165 metabolites had a significantly differential concentration in *ms* comparved with that in CK (Figure 3e and Figure S2). Forty-six categories of metabolites between the *ms* and CK flowers were identified (Figure 5a,b), and 86 metabolites were upregulated and 79 metabolites were downregulated (Figure 5c, Supplementary Materials Data S6). Those metabolites were mainly concentrated in 17 metabolic processes according to the pathway enrichment analysis including flavonoid biosynthesis, biosynthesis of amino acids, lipid metabolism, isoflavonoid biosynthesis and monoterpenoid biosynthesis (Figure 6a). Among the 165 metabolites, seven categories of metabolites (71 metabolites) might be related to plant fertility (Figure 6b). The cis-4-hydroxy-D-proline, leucine and isoleucine contents in the CK were 1.47, 2.57 and 2.17 times higher than that in *ms*, respectively, (*p*-value < 0.001, *p*-value = 0.001 and *p*-value < 0.001, respectively, Table S5). Picolinic acid, maleamic acid, linoelaidic acid, urocanic acid, tartaric acid, etc., were significantly differentially expressed between CK and *ms* mungbeans (*p*-value = 0.020, *p*-value = 0.002, *p*-value = 0.012, *p*-value = 0.047 and *p*-value = 0.003, respectively).

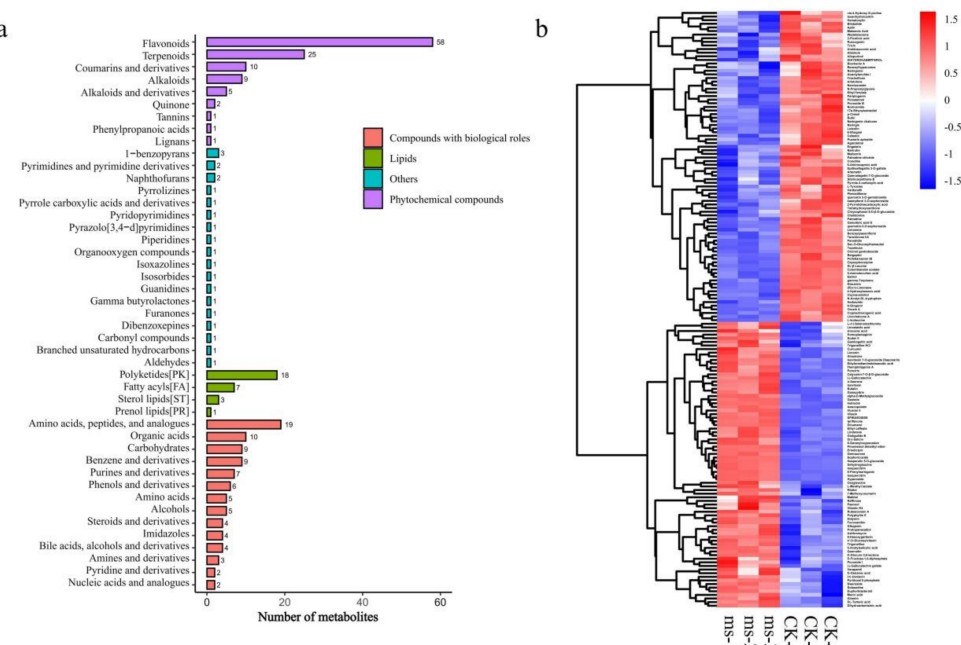

**Figure 5.** Metabolite profiling analyses. The statistical results of 252 metabolites' classification that were identified in mungbean: (**a**) the ordinate represents the metabolic pathway, and the abscissa represents the number of metabolites; (**b**) heatmaps of the differential metabolites' concentration in CK and *ms*.

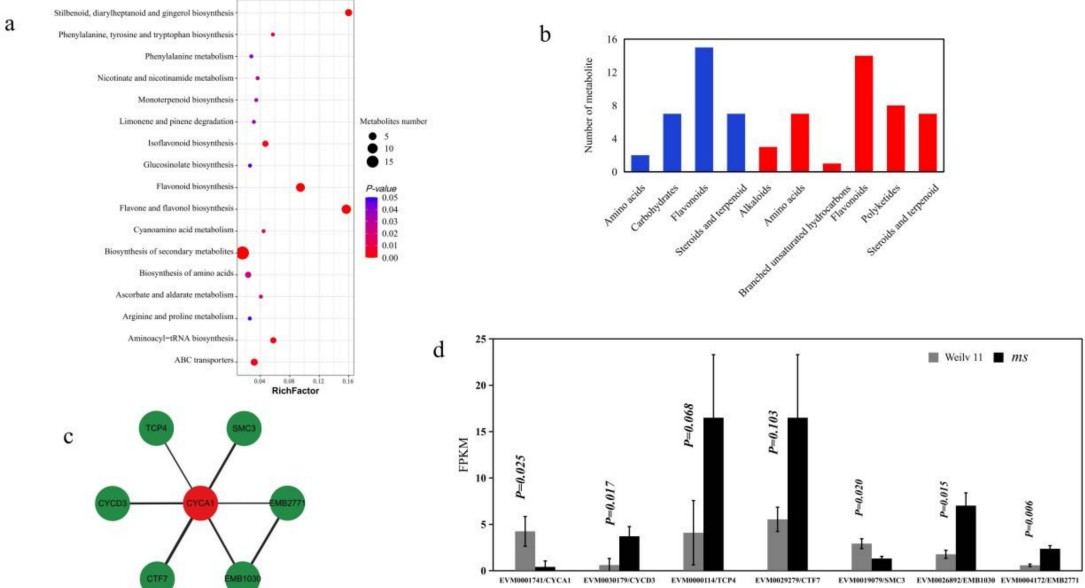

**Figure 6.** Pathway enrichment analysis corresponding to the differential metabolites. Statistical results of 165 significantly differentially concentrated metabolite pathway enrichment analysis. (**a**) The ordinate represents the metabolic pathway, and the abscissa represents the number of metabolites. The coloring indicates the *p*-value, with a higher value in blue and a lower value in red; a lower *p*-value indicates a more significant enrichment. The point size indicates the number of differential metabolites. (**b**) The classification of significantly regulated metabolites; the blue columns represent significantly downregulated metabolites, and the red columns represent significantly upregulated metabolites. (**c**) The protein–protein interaction between *VrCYCA1* and six seed development-related genes. (**d**) Statistical analysis of the seven genes' expression levels in Figure 6c; the *t*-test was used to test the significant differences in gene expression (fpkm) between CK (Weilyu 11) and *ms*.

### 3.5. Genetic Relationships between VrCYCA1, Related Genes and Metabolites in Mungbean

Six genes out of the above 226 DEGs had protein–protein interactions (PPIs) with VrCYCA1 (Figure 6c,d). In addition, six pairs of PPIs were larger than the medium confidence value of 0.40 (Table S6), for example, EVM0001741.1 (VrCYCA1) and EVM0030179.1 (VrCYCD3) (0.77); EVM0001741.1 (VrCYCA1) and EVM0004172.1 (VrEMB2771) (0.67); EVM0001741.1 (VrCYCA1) and EVM0019079.1 (VrSMC3) (0.65). Together with seven categories of metabolites as well as four categories that were up-regulated and six categories that were down-regulated were used to construct the GMS genetic relationships caused by *VrCYCA1* (Figure 7, Table S3). We analyzed the expression levels of the six genes by using the transcriptome data and RT-qPCR experiment, and both of the results showed that only EVM0019079 was downregulated, and the other five genes were upregulated in *ms* compared with that the CK (Figures 6d and S3).

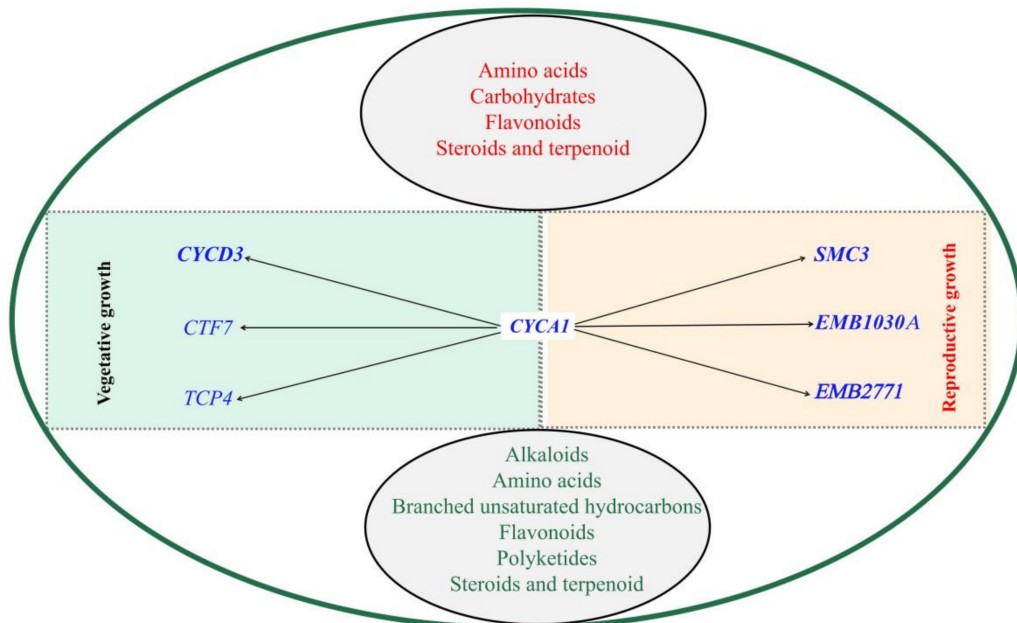

**Figure 7.** The GMS genetic network caused by *VrCYCA1*, related-genes and metabolites in mungbean. Bold genes represent significantly differentially expressed genes. *CYCD3*, *TCP4* and *CTF7* participate in vegetative growth, and *SMC3*, *EMB1030A* and *EMB2771* participate in reproductive growth. The red and green fonts represent the four categories of significantly downregulated metabolites and six categories of significantly upregulated metabolites, respectively.

## 4. Discussion

The utilization of heterosis is one of the most important ways to achieve high-yield improvement in crop breeding. Many GMS-related genes have been discovered in rice, corn and soybean [7,8,13,19]; however, study of mungbean GMS-related genes is limited. As mungbean is a strict self-pollinating crop with a natural outcrossing rate of approximately 1.68% [54], this seriously restricts mungbean hybrid seed production. Therefore, it is important and necessary to mine GMS-related genes in mungbean. In this study, two significant advances were reported. First, the GMS gene, *VrCYCA1*, was firstly reported in mungbean. Then, the GMS genetic network caused by *VrCYCA1* was analyzed according to multi-omics analysis, and the results provide a theoretical basis for quality improvement in mungbean breeding.

### 4.1. GMS in Mungbean

A comprehensive review of the utilization of heterosis in mungbean was presented based on the advances in the authorized patents associated with mungbean heterosis,

discovery of the utilizable mungbean male sterile line and identified candidate genes for GMS would be helpful for mungbean yield improvement.

In this study, the mungbean male sterile mutant, *ms,* was successfully obtained by EMS mutagenesis. The *ms* lines could not normally disperse pollen, and the pollen aborted almost without or with a small pod setting rate (Figure 1b,c). This small amount of selfpods suggest that the female gametes might be fertile or partially fertile and could be stably inherited in the offspring. This self-pod setting might be due to the incomplete pollen abortion of ms or a small amount of normal pollen produced during meiosis.

In our previous study, a mungbean chasmogamous mutant (CM) was identified with petals, a missing keel flap and other exertions, which broke the physical barrier of cross-pollination, and could be used as male parent material in mungbean hybridizing [55]. In the *ms* lines, the segregation ratio of fertile plants to sterile plants in the fertile segregation population was almost 3:1 (Table 1), indicating that the sterile trait was controlled by a single recessive nuclear gene, and this could be used in mungbean hybridizing as a mungbean male sterile line.

*4.2. Significant Changes in VrCYCA1 Expression Led to GMS in Mungbean*

In this study, we conducted two BSA experiments to identify significant QTNs associated with GMS. Candidate genes for GMS were identified in four steps. First, significant QTNs identified both in the (1), (2) and (4) and (1), (3) and (4) pools were considered reliable QTNs. According to the SNP-index and ED-value analysis, significant QTL (Chr5: 6,835,001–6,935,000 bp) were mined (Figure 3a,b). Second, all the genes between the 20 Kb of significant QTL (Chr5: 6,835,001–6,935,000bp) were mined, and nine genes were found in the candidate region (Figure 3c). Then, the genes or their *Arabidopsis* homologous genes, which were annotated with cell differentiation, meiosis, pollen tube growth, pollen tube development, flavonoid biosynthesis and amino acids biosynthesis biological pathways that could cause male sterility in plants, were identified. In this step, only *EVM0001741* and *EVM0022207* complied with the functional annotation analysis, and *EVM0011784*, *EVM0013125*, *EVM0031917* and *EVM0031043* were found with no *Arabidopsis* homologous genes. In the third step, though pollen transcriptome analysis, we found that *EVM0018917*, *EVM0013125* and *EVM0022207* did not express in pollen (Figure 3d), and *EVM0013050*, *VrCYCA1*, EVM0018917 and EVM0017094 had significantly higher expression levels in CK than in *ms* ($p$-value = 0.049, $p$-value = 0.030, $p$-value = 0.012 and $p$-value = 0.028, respectively) (Table S3). Lastly, we conducted RT-qPCR analysis, and the results showed that *EVM0013050* ($p$-value = 0.021), *EVM0001741* ($p$-value = 0.039) and *EVM0017094* ($p$-value = 0.001) were significantly differentially expressed between *ms* and CK flowers (Figure 3e). Though all six gene expressions decreased, *VrCYCA1* expression was severely affected (Figure 3e), and *VrCYCA1* was the only gene annotated with cell differentiation and had significantly different expression levels in CK and *ms*, which proved it to be a major candidate gene in mungbean GMS.

In order to analyze the reason for the differential expression patterns of *VrCYCA1*, we analyzed the DNA sequence variations and found an ATTATA box missing at approximately 4.7 Kb upstream of the *VrCYCA1* gene (Figure 3c). The TATA box is an important element of transcription, and the sequence variation governs the formation of transcriptional complexes and affects reporter gene expression [56]. According to previous reports, point mutation in the TATA box curtails expression of the $H_2A$ histone gene in vivo [57] and *CYCA1* related to sexual reproduction, which participates in meiotic division of *Arabidopsis* but produces the same meiotic products [58,59]. Based on the above results, we choose *VrCYCA1* as our candidate gene. Finally, the GMS genetic network caused by *VrCYCA1* was analyzed according to multi-omics analysis from the perspective of genes and metabolites.

### 4.3. Pollen Tube Growth, Biological Regulation and Pollen Tube Development Genes Were Altered in ms

There were 6653 DEGs that were identified (Figure 4a,b) from the comparative analysis of the *ms* and CK pollen transcriptomes, and 226 DEGs had functions related to flower development, embryo development and seedling development, including *VrCYCA1*, and the functions of these genes were mainly related to flower development, embryo development ending in seed dormancy, seed development, negative regulation of seed germination, seed coat development and seed maturation (Supplementary Materials Data S5). Among the 226 DEGs, only six genes had protein–protein interactions (PPIs) with *VrCYCA1* (Figure 6c,d). Moreover, *CYCD3*, *TCP4* and *CTF7* mainly participated in vegetative growth in *Arabidopsis*. *CYCD3*, encoding a mitotic cyclin protein, regulated cell division and played a crucial role in leaf morphogenesis [60]. *TCP4* was regulated by miR319 and was involve in heterochronic regulation of leaf differentiation [61]. *CTF7* mutation plants exhibited major defects in vegetative growth, and caused infertility [62]. However, *SMC3*, *EMB1030A* and *EMB2771* mainly participated in the reproductive growth of *Arabidopsis*. *SMC3* could regulate nuclear activity during endosperm development [63]. *EMB1030A* mutations in this locus result in embryo lethality [64], and *EMB2771* encodes a subunit of E3 ubiquitin ligase that participates in *Arabidopsis* female gametogenesis and in embryogenesis [65]. The interaction between *VrCYCA1* and the other five genes plays an important role in mungbean male sterility. The results probably revealed the genetic reason for male sterility in mungbean from the perspective of genes.

### 4.4. Amino Acids Biosynthesis, Lipid Metabolism and Flavonoid Biosynthesis Pathways Were Altered in ms

Nutrients mainly include amino acids, fatty acids and carbohydrates, and those metabolites participate in diverse metabolic processes and affect plant phenotype such as male gametophyte development and pollen wall formation [66–68]. The metabolic changes in the *ms* and CK flowers were also analyzed. In this study, there were 46 categories of metabolites, which mainly concentrated in flavonoid biosynthesis, biosynthesis of amino acids and isoflavonoid biosynthesis (Figure 6a,b). During microspore development, amino acids and proteins are important metabolites that provide nutrition and promote pollen development, such as proline, which can be transformed into other amino acids and are required for male gametophyte development in *Arabidopsis* [62–65]. In rice, lipid involved in male fertility, fatty acids and their derivatives are essential building blocks for pollen wall formation [38,68], and flavonoids are also indispensable for complete male fertility; the flavonoids synthesis gene mutant could reduce the germination rate in rice [39].

In this study, cis-4-hydroxy-D-proline, leucine and isoleucine contents were significant higher in CK than that *ms* (Table S5). Some amino acids and lipids, such as picolinic acid, maleamic acid, linoelaidic acid and urocanic acid, also had significantly differential content between CK and *ms* mungbeans (Table S5). As amino acids and lipids are essential building blocks for pollen wall formation [63–65], the results explain the abnormal pollen morphology. Apiin, licochalcone, naringenin chalcone, mulberrin, etc., were significantly differentially expressed between CK and *ms* mungbeans (*p*-value = 0.006, *p*-value = 0.002, *p*-value = 0.001 and *p*-value = 0.010, respectively). The difference in the contents of those metabolites between CK and *ms* are an important reason for the sterility of mungbean caused by *VrCYCA1* and the six related genes (Figure 7). The research on metabolism change probably revealed the male sterility phenotype from the perspective of metabolites and caused the *ms* phenotype in pollen. The correlation between genes and metabolites deserves further study.

### 5. Conclusions

In this study, the mungbean genic male sterility mutant, *ms*, was successfully obtained by EMS mutagenesis. The normal *ms* growth well during the vegetative stage, however the pod setting rate was almost 0 during the reproductive growth stage. The segregation ratio,

BSA, comparative genomics, transcriptome and RT-qPCR analyses indicated that the sterile character was controlled by a single recessive nuclear gene, *VrCYCA1*. Cytological analysis showed that the pollen morphology of *ms* was abnormal, and most of the germination pores were invisible. Moreover, six genes and seven categories of metabolites related to *VrCYCA1* were identified by transcriptome and metabonomics analyses, which were the main genetic and physical reasons for the GMS of *ms* in mungbean.

**Supplementary Materials:** The following supporting information can be downloaded at: https://www.mdpi.com/article/10.3390/agriculture12050686/s1, Figure S1: Gene Ontology (GO) enrichment analysis for up- and down-regulated genes; Figure S2: Volcano plots were used for visualizing differential metabolites between the CK group and the ms group; Figure S3. Real-time PCR analysis of the expression of the six genes with interaction with *EVM0001741* in mungbean; Table S1: BSA mixed-pool sequencing data; Table S2: The list of primers used in this study; Supplementary Material Data S1: 500,315 SNPs detected in the M$_3$ BSA data; Supplementary Material Data S2: 53,529 SNPs detected in the M$_4$ BSA data; Supplementary Material Data S3: RT-qPCR analysis of nine genes around Chr5: 6,835,000 bp~6,935,000 bp interval; Supplementary Material Data S4: 6653 DEGs expressed in *ms* and CK; Supplementary Material Data S5: 226 DEGs related to flower development, embryo development and seedling development; Supplementary Material Data S6: 165 differential metabolites between CK and *ms*.

**Author Contributions:** Conceptualization, X.Y. and X.C.; methodology, J.L. software, J.L. validation, J.L. formal analysis, J.L., Y.L., J.C., C.X., R.W. and Q.Y.; investigation, X.Y. and X.C.; resources, J.L. data curation, J.L. writing—original draft preparation, J.L., X.Y. and X.C. writing—review and editing, J.L. visualization, J.L. supervision, X.Y. and X.C.; project administration, X.Y. and X.C.; funding acquisition, X.Y. and X.C. All authors have read and agreed to the published version of the manuscript.

**Funding:** This research was funded by China Agriculture Research System of MOF and MARA-Food Legumes CARS-08: CARS-08; National Natural Science Foundation of China (31871696): 31871696; Natural Science Foundation of Jiangsu Province (BK20190257): BK20190257; Jiangsu Seed Industry Revitalization Project (JBGS[2021]004):JBGS[2021]004.

**Institutional Review Board Statement:** Not applicable.

**Informed Consent Statement:** Not applicable.

**Data Availability Statement:** Supporting information is available from the Wiley Online Library or from the author.

**Conflicts of Interest:** The authors declare no conflict of interest.

## Abbreviations

| | |
|---|---|
| BSA | Bulked segregate analysis |
| SNP | Single-nucleotide polymorphism |
| QTNs/QTL | Quantitative trait nucleotides/loci |
| EMS | Ethyl methane sulfonate |
| FPKM | Fragments reads per kilobases per million reads |
| PPI | Protein–protein interaction |
| GMS | Genic male sterility |
| LD | Linkage disequilibrium |
| DEGs | Differentially expressed genes |
| *CYCA1* | Cyclin A1 |

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
