# Peer review of "Identification and Clarification of VrCYCA1: A Key Genic Male Sterility-Related Gene in Mungbean by Multi-Omics Analysis"

_agriculture, doi:10.3390/agriculture12050686_

Round 1

Reviewer 1 Report

The authors have addressed some issues pointed out before. I am happy to see that some additions have been made to the discussion. However, after reading the new version of the manuscript I regret to say that I still have found numerous issues that should be addressed in order to consider this paper :

Main issues:

  • A lot of English-related issues remain, I must urge again to the authors that the text should be submitted to more exhaustive revision. In the present version I find hard to believe that a native English speaker had passed some expressions in the text as good ones. I have detailed some of them in my additional comments below, but keep in mind that it is not an exhaustive list
  • Again, I must insist that transcriptome raw data should be uploaded to appropriate public databases and the proper ID access should be detailed in the manuscript
  • Figures 4,5 6 have still a poor resolution, for example metabolite names in 5b are illegible
  • I must insist in my previous recommendation: the expression levels of the genes that form part of the interaction network identified in figure 6d and 7 should be also confirmed by qRT-PCR.

Additional comments:

Regarding Abstract

Line 9: “…in crops...”

Lines 11-12: “by EMS mutagenesis of Weilyu cultivar”

Line 14: “nine genes were found”

Line 18: “in the mutant ms lines compared with the control group”

Lines 18-20: genes have expression levels, metabolites have quantity or concentration levels…hence this sentence should be rewritten accordingly, something like “Moreover, 6653 genes showed differential expression between Weilyu lines and mutant lines, as well as 165 metabolites with significant differences in their concentration levels.”

Lines 20-21: “Amongst those differentially expresses genes, 226 are annotated with functional categories involved in…”

Line 23: “This study has used a multi-omics data approach to identify a mungbean GMS-related gene, as well as a GMS genetic network which can be used to explore the involvement of VrCYCA1 in GMS” (  VrCYCA1 without italics, since it refers to the protein coded by the gene)

Line 34: “a prominent work in crop heterosis utilization” (just to avoid the use of “important“, that has been already used in the prior sentence)

Line 36: “To date, …”

Lines 38-43: Please rectify the commas, which should not appear in italics

Line 43: “These genes are mainly involved in…”

Line 46: “which is required for proper pollen tube orientation”

Line 48: “much less about genes and metabolites involved in GMS process”

Line 49: I'm not quite sure what the authors mean by "clarification of GMS-related genes"

Line 51: “the most commonly used methods are…”

Line 59: “have been identified”

Line 76: “Usually, linkage analysis trends maps a significant locus into a large chromosome interval, …”

Line 89: “At present, …”

Lines 91-92: “mining GMS-related genes and analyzing their related genetic networks will provide…”

Line 93: BSA is already defined in line 52, also “Two BSA data sets”

Lines 99-100: “Moreover, the GMS-related genetic networks were established by…”

Line 104: “Key candidate genes identified in this study”

Regarding Material and methods:

Lines 108-109: Please try to be more accurate in your descriptions: EMS mutagenesis of Weilyu11 seeds?. This Weilyu11 mungbean line is described in the literature? Which is the origin of this mungbean line: commercial, a germplasm bank?

Lines 113-124: I still see the description of the breeding of the methodology of section 2.2 somewhat confusing, perhaps this is a result of a bad translation from the main language of the authors. I insist on my recommendation for review by a native English speaker of the manuscript with experience in this discipline.

- both descendants (fertile and non-fertile) are harvested for each plant?

-Again, I am not very sure how are obtained the M4 lines and particularly about the meaning of “After removed the phenotypic with non-segregated individuals in M2”

Line 123: “Student t-test”

Line 132 “abortion rate of pollen was estimated by observation of their staining rate”

Lines 134-135: “Stalks changes between control and male sterile plants were determined by stalk length measurement”.

Lines 145: “were examined by”

Line 147: “were post-fixed”

Lines 149: OsO4 was removed by rinsing for three times in 0.1 M PBS, 15 min each time”

Lines 153: Sections were allowed to air dry overnight at room temperature. Samples were placed to…”

Line 161: “for DNA extraction”

Line 162: I do not see the relevance of the year of the DNA extraction…

Line 163-164:  I must insist in my comment from my previous review: the detail of N2 freezing is unnecessary an inaccurate…freezing of samples avoids DNA degradation (and other detrimental effects) but also helps with grinding and homogenization. CTAB methodology already includes the use of previously liquid nitrogen frozen samples…so I do not understand why this detail appear so relevant to the authors in this case…is a common routine detail of plant molecular biology techniques regarding DNA extraction

Line 167: BWA software should be referenced

Line 186: “significant QTNs”

Line 187: “were considered”

Line 188: “was chosen”

Lines 190-197: the methodology for the identification of GMS-related genes is not accurate in some cases: in the second step, I think that the correct sequence of events is: identification of DEGs by comparison of ms and CK transcriptomes obtained by RNAseq, and  further selection from the pool of DEG, those genes  annotated with male sterility related functional categories and in a third step, verification of the expression levels of the selected genes by RT-qPCR.

Lines 199-208: the methodology to calculate relative expression levels by qRT-PCR should be detailed or referenced (so, please include a reference of the deltadelta Ct method). Also, statistical methodology and threshold criteria to consider significant changes in gene expression should be detailed. Also, the statistical software used. Tubulin is used as a reference gene, not control gene.

Line 208: what is RCR of tubulin?

Line 230: I don’t think that the use of the expression “were blast” could be the correct term to describe the identification of metabolites area using MMCD and its relative quantification between samples by comparing peak areas.

Line 231: threshold criteria to consider significant changes in metabolite levels should be detailed, also the statistical software used.

Regarding Results:

Line 250: “genetic”

Line 262: statistic test used to test significant differences in flower stalks length between Weilyu 11 and ms plants should be detailed in figure 1 legend

Lines 266-267: please check the sentence “and the number of the pollen was larger than that in the CK_”….I am confused… CK plants are not wild type?

Lines 285-286: “…genes, two pools consisting of …”

Lines 287-288: “ In total, 196869736, 213597502, 193903636 and 200865064  …”

Lines 291-292: “500315 and 53529 SNPs”

Lines 324-329:  The results described in lines 300-306 are repeated here…why?

Line 362: “seed” is repeated, please correct  (also in supplementary table S3 legend)

Regarding discussion:

Line 394: “yield improvement”

Lines 428-435: the description results regarding  VrCYCA1 expression levels are repeated two times.

Line 462: “which participates”

Lines 463-465: In the response to my preovous comments the authors mention that “discussion of the proteins encoded for the different genes detected, as well as the metabolites and pathways was conservative”. In my opinion the sentences “the results revealed the genetic reason of male sterility in mungbean from the perspective of genes “ is far away from being “conservative” , because the authors have not provided any other evidences about the involvement of this gene network in GMS in mungbean.

Lines 483, 487-489: From my previous review,  I must insist that metabolites are not expressed. Metabolites have different levels of abundance, differential content….

Lines 489-493: Again the sentences “The difference contents of those metabolites between CK and ms are important reason for the sterility of mungbean caused by VrCYCA1 and six related genes” or  “ The research on metablism change revealed the physical reason of male sterility in mungbean from the perspective of metabolites, and caused the ms phenotype in pollen” is in my opinion far from conservative: the authors have not provides any evidence of the direct involvement of these metabolites in GMS

Line 491: “metabolism change”

Author Response

Respond to the first reviewer

Main issues:

Question 1 A lot of English-related issues remain, I must urge again to the authors that the text should be submitted to more exhaustive revision. In the present version I find hard to believe that a native English speaker had passed some expressions in the text as good ones. I have detailed some of them in my additional comments below, but keep in mind that it is not an exhaustive list.

Answer: Thanks very much for your comments and suggestions and in the revision we have addressed this issue as can as possible in the revision. And the manuscript were carefully checked thorough language editing by a professional before resubmit. For us, language issues have always been our weakness, however we will try to improve the problem.

Question 2 Again, I must insist that transcriptome raw data should be uploaded to appropriate public databases and the proper ID access should be detailed in the manuscript.

Answer: Thanks very much for your suggestions, the transcriptome raw data was uploaded to NCBI and the accession was PRJNA822679, and the IDwas 822679. We addressed the question in the manuscript in line 272-278.

Question 3 Figures 4,5 6 have still a poor resolution, for example metabolite names in 5b are illegible

Answer: Thanks very much for your suggestions, except the PDF file, we revised figures 4,5 6 with a more high resolution.

Question 4 I must insist in my previous recommendation: the expression levels of the genes that form part of the interaction network identified in figure 6d and 7 should be also confirmed by qRT-PCR.

Answer: Thanks very much for your suggestions, the expression of genes in figure 6d and 7 were confirmed by qRT-PCR in the revision, and the result were showed in line 395-398.

Additional comments:

Line 9: “…in crops...”

Answer: Thanks very much for your comments and we have revised the related sentences in the revision (lines 8).

Lines 11-12: “by EMS mutagenesis of Weilyu cultivar”

Answer: Thanks very much for your comments and we have revised the related sentences in the revision (lines 11-12).

Line 14: “nine genes were found”

Answer: Thanks very much for your comments and we have revised the related sentences in the revision (lines 14).

Line 18: “in the mutant ms lines compared with the control group”

Answer: Thanks very much for your comments and we have revised the related sentences in the revision (lines 17).

Lines 18-20: genes have expression levels, metabolites have quantity or concentration levels…hence this sentence should be rewritten accordingly, something like “Moreover, 6653 genes showed differential expression between Weilyu lines and mutant lines, as well as 165 metabolites with significant differences in their concentration levels.”

Answer: Thanks very much for your comments and we have revised the related sentences in the revision (lines 17-19).

Lines 20-21: “Amongst those differentially expresses genes, 226 are annotated with functional categories involved in…”

Answer: Thanks very much for your comments and we have revised the related sentences in the revision (lines 19-20).

Line 23: “This study has used a multi-omics data approach to identify a mungbean GMS-related gene, as well as a GMS genetic network which can be used to explore the involvement of VrCYCA1 in GMS” (VrCYCA1 without italics, since it refers to the protein coded by the gene)

Answer: Thanks very much for your comments and we have revised the related sentences in the revision (lines 19-20).

Line 34: “a prominent work in crop heterosis utilization” (just to avoid the use of “important“, that has been already used in the prior sentence)

Answer: Thanks very much for your comments and we have revised the related sentences in the revision (lines 34-35).

Line 36: “To date, …”

Answer: Thanks very much for your comments and we have revised the related sentences in the revision (lines 37).

Lines 38-43: Please rectify the commas, which should not appear in italics

Answer: Thanks very much for your comments and we have revised the related sentences in the revision (lines 37-42).

Line 43: “These genes are mainly involved in…”

Answer: Thanks very much for your comments and we have revised the related sentences in the revision (lines 44).

Line 46: “which is required for proper pollen tube orientation”

Answer: Thanks very much for your comments and we have revised the related sentences in the revision (lines 47).

Line 48: “much less about genes and metabolites involved in GMS process”

Answer: Thanks very much for your comments and we have revised the related sentences in the revision (lines 49).

Line 49: I'm not quite sure what the authors mean by "clarification of GMS-related genes"

Answer: Thanks very much for your comments and we have revised the related sentences in the revision (lines 51).

Line 51: “the most commonly used methods are…”

Answer: Thanks very much for your comments and we have revised the related sentences in the revision (lines 52).

Line 59: “have been identified”

Answer: Thanks very much for your comments and we have revised the related sentences in the revision (lines 61).

Line 76: “Usually, linkage analysis trends maps a significant locus into a large chromosome interval, …”

Answer: Thanks very much for your comments and we have revised the related sentences in the revision (lines 77-78).

Line 89: “At present, …”

Answer: Thanks very much for your comments and we have revised the related sentences in the revision (lines 91).

Lines 91-92: “mining GMS-related genes and analyzing their related genetic networks will provide…”

Answer: Thanks very much for your comments and we have revised the related sentences in the revision (lines 92).

Line 93: BSA is already defined in line 52, also “Two BSA data sets” 

Answer: Thanks very much for your comments and we have revised the related sentences in the revision (lines 95).

Lines 99-100: “Moreover, the GMS-related genetic networks were established by…”

Answer: Thanks very much for your comments and we have revised the related sentences in the revision (lines 102).

Line 104: “Key candidate genes identified in this study”

Answer: Thanks very much for your comments and we have revised the related sentences in the revision (lines 106).

Regarding Material and methods:

Lines 108-109: Please try to be more accurate in your descriptions: EMS mutagenesis of Weilyu11 seeds?. This Weilyu11 mungbean line is described in the literature? Which is the origin of this mungbean line: commercial, a germplasm bank?

Answer: Thanks very much for your comments and we have revised the related sentences in the revision (lines 112).

Lines 113-124: I still see the description of the breeding of the methodology of section 2.2 somewhat confusing, perhaps this is a result of a bad translation from the main language of the authors. I insist on my recommendation for review by a native English speaker of the manuscript with experience in this discipline. 

Answer: Thanks very much for your comments and we have revised the related sentences in the revision (lines 119-122).

- both descendants (fertile and non-fertile) are harvested for each plant?

Answer: Thanks very much for your comments and fertile and non-fertile are harvested for each plant from M3.

-Again, I am not very sure how are obtained the M4 lines and particularly about the meaning of “After removed the phenotypic with non-segregated individuals in M2

Answer: Thanks very much for your comments and M3 lines fertile and non-fertile are harvested, M4 lines were obtained from M3 (fertile and non-fertile) lines. “After removed the phenotypic with non-segregated individuals in M2 means” only fertile lines in M2 were saved.

Line 123: “Student t-test”

Answer: Thanks very much for your comments and we have revised the related sentences in the revision (lines 126).

J Tarasińska. (2005). Confidence intervals for the power of student's t-test. Statistics & Probability Letters, 73(2), 125-130.

Line 132 “abortion rate of pollen was estimated by observation of their staining rate”

Answer: Thanks very much for your comments and we have revised the related sentences in the revision (lines 136).

Lines 134-135: “Stalks changes between control and male sterile plants were determined by stalk length measurement”.

Answer: Thanks very much for your comments and we have revised the related sentences in the revision (lines 138-139).

Lines 145: “were examined by”

Answer: Thanks very much for your comments and we have revised the related sentences in the revision (lines 149).

Line 147: “were post-fixed”

Answer: Thanks very much for your comments and we have revised the related sentences in the revision (lines 152).

Lines 149: OsO4 was removed by rinsing for three times in 0.1 M PBS, 15 min each time”

Answer: Thanks very much for your comments and we have revised the related sentences in the revision (lines 154-155).

Lines 153: Sections were allowed to air dry overnight at room temperature. Samples were placed to…”

Answer: Thanks very much for your comments and we have revised the related sentences in the revision (lines 158).

Line 161: “for DNA extraction”

Answer: Thanks very much for your comments and we have revised the related sentences in the revision (lines 166).

Line 162: I do not see the relevance of the year of the DNA extraction…

Answer: Thanks very much for your comments and we have revised the related sentences in the revision (lines 166).

Line 163-164: I must insist in my comment from my previous review: the detail of N2 freezing is unnecessary an inaccurate…freezing of samples avoids DNA degradation (and other detrimental effects) but also helps with grinding and homogenization. CTAB methodology already includes the use of previously liquid nitrogen frozen samples…so I do not understand why this detail appear so relevant to the authors in this case…is a common routine detail of plant molecular biology techniques regarding DNA extraction

Answer: Thanks very much for your comments and we have revised the related sentences in the revision (lines 167-168).

Line 167: BWA software should be referenced

Answer: Thanks very much for your comments and we have revised the related sentences in the revision (lines 170).

Line 186: “significant QTNs”

Answer: Thanks very much for your comments and we have revised the related sentences in the revision (lines 185).

Line 187: “were considered”

Answer: Thanks very much for your comments and we have revised the related sentences in the revision (lines 192).

Line 188: “was chosen”

Answer: Thanks very much for your comments and we have revised the related sentences in the revision (lines 193).

Lines 190-197: the methodology for the identification of GMS-related genes is not accurate in some cases: in the second step, I think that the correct sequence of events is: identification of DEGs by comparison of ms and CK transcriptomes obtained by RNAseq, and further selection from the pool of DEG, those genes annotated with male sterility related functional categories and in a third step, verification of the expression levels of the selected genes by RT-qPCR.

Answer: Thanks very much for your comments and suggestion. That's a good question, in the candidate genes identification, candidate genes refered to genes linked to the significant QTN. I dentification of DEGs by comparison of ms and CK transcriptomes obtained by RNAseq, those DEGs were with unknown source. (lines 193).

Lines 199-208: the methodology to calculate relative expression levels by qRT-PCR should be detailed or referenced (so, please include a reference of the deltadelta Ct method). Also, statistical methodology and threshold criteria to consider significant changes in gene expression should be detailed. Also, the statistical software used. Tubulin is used as a reference gene, not control gene.

Answer: Thanks very much for your comments and we have revised the related sentences in the revision (lines 210-214).

Line 208: what is RCR of tubulin?

Answer: Thanks very much for your comments and we have revised the related sentences in the revision (lines 216).

Line 230: I don’t think that the use of the expression “were blast” could be the correct term to describe the identification of metabolites area using MMCD and its relative quantification between samples by comparing peak areas.

Answer: Thanks very much for your carefulness and we have revised the related sentences in the revision (lines 237-238).

Line 231: threshold criteria to consider significant changes in metabolite levels should be detailed, also the statistical software used.

Answer: Thanks very much for your carefulness and we have revised the related sentences in the revision (lines 240-241).

Regarding Results:

Line 250: “genetic”

Answer: Thanks very much for your carefulness and we have revised the related sentences in the revision (lines 249).

Line 262: statistic test used to test significant differences in flower stalks length between Weilyu 11 and ms plants should be detailed in figure 1 legend

Answer: Thanks very much for your carefulness and we have revised the related sentences in the revision (lines 276-277).

Lines 266-267: please check the sentence “and the number of the pollen was larger than that in the CK_”….I am confused… CK plants are not wild type?

Answer: Thanks very much for your carefulness and we have revised the related sentences in the revision (lines 282).

Lines 285-286: “…genes, two pools consisting of …”

Answer: Thanks very much for your carefulness and we have revised the related sentences in the revision (lines 300-301).

Lines 287-288: “ In total, 196869736, 213597502, 193903636 and 200865064  …”

Answer: Thanks very much for your suggestions and we have revised the related sentences in the revision (lines 301).

Lines 291-292: “500315 and 53529 SNPs”

Answer: Thanks very much for your suggestions and we have revised the related sentences in the revision (lines 302).

Lines 324-329: The results described in lines 300-306 are repeated here…why?

Answer: Thanks very much for your carefulness and we have revised the related sentences in the revision were deleted (lines 318-319).

Line 362: “seed” is repeated, please correct  (also in supplementary table S3 legend)

Answer: Thanks very much for your carefulness and we have revised the related sentences in the revision were deleted (lines 384).

Regarding discussion:

Line 394: “yield improvement”

Answer:Thanks very much for your suggestions and we have revised the related sentences in the revision (lines 408).

Lines 428-435: the description results regarding VrCYCA1 expression levels are repeated two times.

Answer: Thanks very much for your carefulness and we have deleted related sentences in the revision were deleted (lines 460).

Line 462: “which participates”

Answer:Thanks very much for your suggestions and we have revised the related sentences in the revision (lines 467).

Lines 463-465: In the response to my preovous comments the authors mention that “discussion of the proteins encoded for the different genes detected, as well as the metabolites and pathways was conservative”. In my opinion the sentences “the results revealed the genetic reason of male sterility in mungbean from the perspective of genes “ is far away from being “conservative” , because the authors have not provided any other evidences about the involvement of this gene network in GMS in mungbean.

Answer:Thanks very much for your suggestions and in the discussion we have used the genes and metabolites to explain the possible reasons, that caused mungbean GMS, which could provides a good direction for follow-up research. And, we revised the related sentences in the revision (lines 491 and lines 497-498 ).

Lines 483, 487-489: From my previous review,  I must insist that metabolites are not expressed. Metabolites have different levels of abundance, differential content….

Answer: Thanks very much for your carefulness and we have revised the related sentences (lines 491 and lines 511).

Lines 489-493: Again the sentences “The difference contents of those metabolites between CK and ms are important reason for the sterility of mungbean caused by VrCYCA1 and six related genes” or  “ The research on metablism change revealed the physical reason of male sterility in mungbean from the perspective of metabolites, and caused the ms phenotype in pollen” is in my opinion far from conservative: the authors have not provides any evidence of the direct involvement of these metabolites in GMS

Answer:Thanks very much for your suggestions. And, we revised the related sentences in the revision (lines 518-519). And, the molecular function of VrCYCA1 will continue.

Line 491: “metabolism change”

Answer: Thanks very much for your carefulness and we have revised the related sentences (lines 518).

Author Response

Respond to the second reviewer

Question 1 line 18: “compared” 

Answer: Thanks very much for your comments and we have revised the related sentences in the revision (lines 17).

Question 2 line 23: “to mine” 

Answer: Thanks very much for your comments and we have revised the related sentences in the revision (lines 24).

Question 3 line 114-119: “Paragraph is not clear” 

Answer: Thanks very much for your comments and we have revised the related sentences in the revision (lines 119-122).

Question 4 line 244-245: “Paragraph is not clear” 

Answer: Thanks very much for your comments and we have revised the related sentences in the revision (lines 245).

Question 5 line 303: “conducted” 

Answer: Thanks very much for your comments and we have revised the related sentences in the revision (lines 338).

Reviewer 3 Report

The authors' revision have improved the paper but there are still some remaining edits required. The abstract and introduction now start with a clearer link between heterosis and male sterility providing better context for the study. More acronyms have been explained to help the reader but a few remain. The study aims are clearer to understand. The methods are now more complete with references as required and the methods match better with the presented results. Several edits have been made to the results and discussion that improve their presentation but a few issues remain as described in the specific comments. Metabolite results are still incorrectly presented in the discussion section. The conclusion that metabolite differences cause male sterility is too strong as you have only observed an association. Additional edits have improved the language and readability of the text. 

Specific comments
L18 Typo: change "comparved" to "compared"
L61 Explain BSA acronym.
L76 Typo: change "mapps" to "mapped"
L100 Typo: change "establishes" to "established"
L164-165 There is still a gap in the methods from DNA extraction to sequencing library construction. If this step was done by a sequencing service, the service provider should be referenced here.
L202 Should "progress" be "process"? 
L208 Typo: change "RCR" to "PCR"
L277-283 This section is wrongly formatted as main text and is part of the legend of figure 2. Typo: change "cotyledon" to "anther"
L297 Test related to p value result is not provided.
Figure 4 The labels of this figure remain very small.
L344 No vertical lines shown in this figure panel.
L402 The barrier is physical not physiological in the chasmogamous flower mutant.
L419 Typo: change "compliance" to "complied".
L444 Change "And" to "Finally" to improve readability.
L448-452 Several gene functions are repeated twice.
L472 Typo: Remove "and".
L480-489 Metabolite results should be moved to the results section.
L492 I disagree that you have shown that these metabolite differences cause male sterility. The observed differences are associated with the male sterility phenotype. 

Author Response

Respond to the third reviewer

Comments and Suggestions for Authors

The authors' revision have improved the paper but there are still some remaining edits required. The abstract and introduction now start with a clearer link between heterosis and male sterility providing better context for the study. More acronyms have been explained to help the reader but a few remain. The study aims are clearer to understand. The methods are now more complete with references as required and the methods match better with the presented results. Several edits have been made to the results and discussion that improve their presentation but a few issues remain as described in the specific comments. Metabolite results are still incorrectly presented in the discussion section. The conclusion that metabolite differences cause male sterility is too strong as you have only observed an association. Additional edits have improved the language and readability of the text.

Specific comments
Question 1 L18 Typo: change "comparved" to "compared"

Answer: Thanks very much for your comments and we have revised the related sentences in the revision (lines 17).

Question 2 L61 Explain BSA acronym.

Answer: Thanks very much for your comments and we have revised the related sentences in the revision (lines 53). Here, BSA was abbreviation.

Question 3 L76 Typo: change "mapps" to "mapped"

Answer: Thanks very much for your comments and we have revised the related sentences in the revision (lines 77).

Question 4 L100 Typo: change "establishes" to "established"

Answer: Thanks very much for your comments and we have revised the related sentences in the revision (lines 102).

Question 5 L164-165 There is still a gap in the methods from DNA extraction to sequencing library construction. If this step was done by a sequencing service, the service provider should be referenced here.

Answer: Thanks very much for your comments and we have revised the related sentences in the revision (lines 167-168). And, added a reference.

Question 6 L202 Should "progress" be "process"?

Answer: Thanks very much for your carefulness and here "progress" is "parameter" (lines 208).

Question 7 L208 Typo: change "RCR" to "PCR"

Answer: Thanks very much for your carefulness (lines 216).

Question 8 L277-283 This section is wrongly formatted as main text and is part of the legend of figure 2. Typo: change "cotyledon" to "anther"

Answer: Thanks very much for your carefulness and we have revised the related sentences in the revision (lines 293-298).

Question 9 L297 Test related to p value result is not provided.

Answer: Thanks very much for your carefulness and we have revised the related sentences in the revision (lines 312).

Question 10 Figure 4 The labels of this figure remain very small.

Answer: Thanks very much for your suggestions and we have revised the related figures in the revision.

Question 11 L344 No vertical lines shown in this figure panel.

Answer: Thanks very much for your carefulness and we have revised figure 4.

Question 12 L402 The barrier is physical not physiological in the chasmogamous flower mutant.

Answer: Thanks very much for your carefulness and we have revised the related sentences in the revision (lines 430).

Question 13 L419 Typo: change "compliance" to "complied". 435

Answer: Thanks very much for your carefulness and we have revised the related sentences in the revision (lines 447).

Question 14 L444 Change "And" to "Finally" to improve readability. 460

Answer: Thanks very much for your suggestions and we have revised the related sentences in the revision (lines 469).

Question 15 L448-452 Several gene functions are repeated twice.

Answer: Thanks very much for your carefulness and we have detected the related sentences in the revision (lines 460).

Question 16 L472 Typo: Remove "and". 488

Answer: Thanks very much for your suggestions and we have revised the related sentences in the revision (lines 499).

Question 17 L480-489 Metabolite results should be moved to the results section.

Answer: Thanks very much for your suggestions and we have revised the related sentences in the revision (lines 352-358 and lines 508-5012).

Question 18 L492 I disagree that you have shown that these metabolite differences cause male sterility. The observed differences are associated with the male sterility phenotype.

Answer: Thanks very much for your suggestions and we have revised the related sentences in the revision (lines 518-519).

Round 2

Reviewer 1 Report

The main issues pointed previously have been addressed by the authors. Still, I feel that several minor issues should be addressed:

Regarding introduction

Line 42: “MAC1、ms9、Ocl4、Ms7、ms26、ms30、ipe2” commas should not be in italics, also please delete the comma after ipe2

Lines 49-50: “identification of GMS-related genes”

Regarding material and methods

Line 111: I must insist… This Weilyu11 mungbean line is described in the literature? Which is the origin of this mungbean line: commercial, a germplasm bank? These details should appear.

Line 129: t-test reference detailed by the authors should appear here (Tarasińska 2005)

Line 210: according to the reference list: deltadelta ct method should be [48] instead of [47]. Please double check the rest of your references

Line 239: Again, I must insist that I don’t think that the use of the expression “were blast” could be the correct term to describe the identification of metabolites area using MMCD and its relative quantification between samples by comparing peak areas.

Line 248: “genetic”

Regarding Discussion

Lines 405-406: “high yield improvement in crop breeding”

Line 470: “Finally, …”

Regarding Supplementary material

Figure S3: qRT-PCR figure should include statistical analysis.

Figure S3 legend: “Real-time PCR analysis of the expression of the six genes with interaction with EVM0001741 in mungbean”

Author Response

Respond to the first reviewer

Comments and Suggestions for Authors

The main issues pointed previously have been addressed by the authors. Still, I feel that several minor issues should be addressed:

Regarding introduction

Question 1 Line 42: “MAC1、ms9、Ocl4、Ms7、ms26、ms30、ipe2” commas should not be in italics, also please delete the comma after ipe2

Answer: Thanks very much for your carefulness and we have revised the related sentences in the revision (line 43).

Question 2 Lines 49-50: “identification of GMS-related genes”

Answer: Thanks very much for your carefulness and we have revised the related sentences in the revision (line 51).

Regarding material and methods

Question 3 Line 111: I must insist… This Weilyu11 mungbean line is described in the literature? Which is the origin of this mungbean line: commercial, a germplasm bank? These details should appear.

Answer: Thanks very much for your comments and Weilyu11 was a cultivar accession saved in Institute of Industrial Crops, Jiangsu Academy of Agricultural Sciences, if need we can offer the seeds for scientific research. Weilyu1

Question 4 Line 129: t-test reference detailed by the authors should appear here (Tarasińska 2005)

Answer: Thanks very much for your comments and we have revised the related sentences in the revision (lines 129 and 648).

Question 5 Line 210: according to the reference list: deltadelta ct method should be [48] instead of [47]. Please double check the rest of your references

Answer: Thanks very much for your carefulness and we have revised the related references in the revision (lines 652-691).

Question 6 Line 239: Again, I must insist that I don’t think that the use of the expression “were blast” could be the correct term to describe the identification of metabolites area using MMCD and its relative quantification between samples by comparing peak areas.

Answer: Thanks very much for your comments and we used the measurements mention in the references blow. 

  1. Liu, J.Y.; Zhang, Y.W.; Han, X.; Zuo, J.F.; Zhang, Z.; Shang, H.; Song, Q.; Zhang, Y.M. An evolutionary population structure model reveals pleiotropic effects of GmPDATfor traits related to seed size and oil content in soybean. J Exp Bot 2020, 71, 6988-7002.
  2. Cui, Q.; Lewis, IA.; Hegeman, AD.; Anderson, M.E.; Li, J.; Schulte, C.F.; Westler, W.M.; Eghbalnia, H.R.; Sussman, M.R.; Markley J.L. Metabolite identification via the Madison Metabolomics Consortium Database. Nat Biotechnol 2008, 26, 162-4.
  3. Aharoni, A., Ric de Vos, C. H., Verhoeven, H. A., Maliepaard, C. A., Kruppa, G., Bino, R., & Goodenowe, D. B. Nontargeted metabolome analysis by use of Fourier Transform Ion Cyclotron Mass Spectrometry. Omics 2002, 6(3), 217–234.

Question 7 Line 248: “genetic”

Answer: Thanks very much for your carefulness and we have revised the related references in the revision (line 250).

Regarding Discussion

Question 8 Lines 405-406: “high yield improvement in crop breeding”

Answer: Thanks very much for your carefulness and we have revised the related references in the revision (line 409).

Question 9 Line 470: “Finally, …”

Answer: Thanks very much for your carefulness and we have revised the related references in the revision (line 470).

Regarding Supplementary material

Question 10 Figure S3: qRT-PCR figure should include statistical analysis.

Answer: Thanks very much for your carefulness and we have revised figure S3 and added statistical analysis.

Question 11 Figure S3 legend: “Real-time PCR analysis of the expression of the six genes with interaction with EVM0001741 in mungbean”

Answer:Thanks very much for your suggestions and we have revised the figure legend of figure S3 (lines 549-550).

This manuscript is a resubmission of an earlier submission. The following is a list of the peer review reports and author responses from that submission.

Round 1

Reviewer 1 Report

Some corrections are needed that are mentioned in the attached file below. 

Author Response

Respond to the first reviewer

Question 1 Abstract: Line No: 9: Please ad one more sentence about the background that represent the current study before starting “Although crop fertility” etc.

Answer: Thanks very much for your comments and we have add the related sentences in the revision (lines 8).

Question 2 Abstract: Line No: 14-17: There is repetition of word “and” in single sentence as it should be in accurate form.

Answer: Thanks very much for your comments and we have fixed them and revised the related sentences in the revision (lines 14).

Question 3 Introduction: Line No:28: Author(s) have described nuclear genes in correlation to Genic male sterility(GMS)that is an important trait to study crop heterosis. What types of nuclear genes involved?

Answer: Thanks very much for your comments and we have fixed them and revised the related sentences in the revision (lines 14).

Question 4 Introduction: Line No: 27-29: Mining plant GMS genes has become an important work of crop heterosis utilization. Please ad at least 2 two sentences about the role of Mining plant GMS genes.

Answer: Thanks very much for your comments and we have fixed them and revised the related sentences in the revision (lines 28-31).

Question 5 Introduction: Line No: 72-73: Metabolites are the basis of agronomic traits, and act as a bridge between traits and genes (Fiehn, 2002). What type of traits and genes?. Secondly, Citation is in wrong format. Please follow the journal instructions for references.

Answer: Thanks very much for your comments and we have fixed them and revised the related sentences in the revision (lines77-78).

Question 6 Introduction: Line No:57-64: Some of the Citations in wrong format. Please follow the journal instructions for references.

Answer: Thanks very much for your suggestions. We checked in carefully, and we have revised the related sentences in the revision (lines71-76).

Question 7 Materials and methods: Line No: 102-104: Poorly merged with statistically section. The number of fertile and sterile individuals in the segregated population was counted, and the conformity was statistically analyzed using chi-square test, with p < 0.05 etc. Please mention it separately in statistical analysis section.

Answer: Thanks very much for your suggestions. We have revised the related sentences in the revision (lines111-113).

Question 8 Discussion: Line No: 344; Heterosis utilization in mung bean?. Is this explained as major heading or subheading?

Answer: Thanks very much for your suggestions. We checked in carefully, and we have revised the related sentences in the revision (lines362-366). And related sentences in lines 411-414 and line 433.

Question 9 Discussion: Line No: 384: Please italic scientific term

Answer: Thanks very much for your suggestions. We checked in carefully, and we have revised the related sentences in the revision (lines372 and line 382)

Question 10 Discussion: Line No: 417-418: In rice, lipid involved in male fertility, fatty acids and their derivatives disrupting the pollen wall formation- Is any other amino acid/chemical agents block it?

Answer: Thanks very much for your suggestions. We checked in carefully, it's reported that sporopollenin and tryphine transport-related GMS genes have conserved functions in controlling pollen wall, proper exine formation and anther development across plant species, whereas their actual functions remain to be further proved in rice.

Hu, Y.; Wu, Q.; Liu, S.; Wei, L.; Chen, X.; Yan, Z.; Yu, J.; Zeng, L.; Ding, Y. Study of rice pollen grains by multispectral imaging microscopy. Microsc Res Tech 2005, 15, 335-46.

Wan, X.; Wu, S.; Li, Z.; An, X.; Tian, Y. Lipid Metabolism: Critical Roles in Male Fertility and Other Aspects of Reproductive Development in Plants. Mol Plant 2020, 6, 955-983.

Reviewer 2 Report

Manuscript entitled “Identification and clarification of VrCYCA1, a key genic male sterility related gene in mungbean by multi-omics analysis” by Liu et al describes a comparative study between EMS mutagenized-lines of mungbean with male-sterility phenotype (ms lines),  and wild-type ones (CK). Their phenotype was compared by several approximations (pollen viability, citologila analysis of flower and pollen morphology), showing pollen. Also, the authors show that male-sterility trait segregation of ms lines is compatible with a recessive mutation on a single nuclear gene. The mutated gene is subsequently identified through a QTL study comparing sets of male-sterile and fertile plants. The authors show that the ms lines present a point mutation on the promoter region of VrCYCA1, which prevents its expression. The effects of the lack of VrCYCA1 expression are further examined in comparative studies of transcriptomes and metabolomes of flowers of MS and CK lines, where sets of genes and metabolites with differential expression or levels were identified. The authors perform enrichment analyses of the functional categories and pathways associated with these genes and metabolites in order to identify possible mechanisms responsible for the male sterility phenotype of VrCYCA1 mutant plants. The results obtained are interesting and add new data about the regulatory mechanisms implicated in male sterility in mungbean. The experimental design and the proposed approaches used in this work seem both correct for the most part. However, in my opinion, some aspects of the manuscript need to be revised before considering this work suitable for publication. Some of them are enumerated in the additional comments below:

Major concerns:

  1. One of the main aspects that weighs down this version of the manuscript is the grammar and use of English. I have pointed out some suggestions in my additional comments below, but I strongly recommend a more exhaustive revision by a native English speaker.
  2. Transcriptome and metabolome data should be uploaded to appropriate public databases.
  3. Figures 3,4,5: image resolution is very poor: the text is illegible in most cases
  4. the discussion of work in this version of the manuscript is often rather superficial and should be improved. The authors are too inclined to repeat the description of the results again, and sometimes the authors stop in this point, instead of integrating them into previous results from the literature that helps to arrive at some new interpretation regarding the observed results. In this sense, I must make the authors two recommendations:
    1.  to to place their results in the context of a broader framework of the previous literature.
    2.  the sets of genes and metabolites detected in the different comparisons are also discussed in a very superficial way.  The authors should try to further discuss the biological implications of the proteins encoded for the different genes detected, as well as the metabolites and pathways.
  5. Statistical methodology and threshold criteria to consider significant changes is absent for gene expression and metabolite analyses is absent in material and methods section
  6. Expression levels of the genes that form part of the interaction network identified in (figure 7c and yd) should be also confirmed by qRT-PCR. By the other side, some expression changes of some genes (TCP4, CTF7) should not be described as significant (their P> 0,05)

For all these reasons, I consider that the present manuscript is still far from enough, in my opinion, to consider its publication in this journal.

Additional comments.

  • Regarding Introduction

Line 10: GMS should be defined

Lines 14-16: Please consider rephrase and join the two sentences regarding expression of VrCYCA1 gene

Lines 17-18: metabolites are synthetized or degraded but not “expressed”

Lines 18-120: genes are segments of DNA that code for proteins, these proteins perform tasks. Hence the genes are not involved in anything of had interactions with other genes…. please rephrase accordingly

Lines 21-2 please check grammar of the sentence

Line 28: “Mining plant GMS-related genes “ I suggest that the authors make this modification when referring to GMS-relates genes as in lines 31, 33 and so on

Lines 37, 39: again, these genes code for proteins involved in these processes…please modify the sentence accordingly

Lines 41: “much less about…”!

Line 48 “GWAS is generally applicable”

Line 50: BC, DH, RIL or NAM abbreviations have not been defined

Lines 51-52: please check the use of verbal tenses in” To date, thousands of reported QTN/QTL involved in 51 quantitative traits, have been identified by GWAS and linkage analysis”.

Line 64: MER3 should appear in italics

Line 63: “…used this method to identify…”

Line 66-67: I don’t understand why “analysis trends” and large genetic interval” appear in italics

Line 68: in this case (Liu et al 2020a) should be removed since is already references with “[32]”

Lines 68-70 please include a brief description ot the work of liut el at 2020a to inform to the reader at least about which plant/process t in particular are studying these authors.

Lines 73-74: Again, “(Fiehn 202)” is already referenced by [35]

Lines 77-80: Please check grammar od the sentence

Lines 81 and 247: I am not very sure that “consists” should be used in these sentences

Line 86-87: “were establishes by identification of differentially-expressed genes”

Line 88: “According to the results obtained”

  • Regarding Materials and methods

Line 94: the origin of “ms” lines should be detailed or referenced. Were these EMS-mutagenized lines generated in this work or have they already been described in a previous manuscript?

Line 95: Please check “ofin”

Lines 99-100: I think that the methodology of section 2.2 should be detailed more accurately what means “After removed the 99 phenotypic with non-segregated individuals,”

Line: 106: Please use the appropriate verbal tenses: “flower buds and blooming flowers of “Weilyu11” and ms mutants were collected at different development stages during….”

Line 114: DAF abbreviation should be defined

Line 116: FAA abbreviation should be defined

Line 124: PBS abbreviation should be defined

Lines 126-127: please use mora appropriate verbal tenses

Lines 130-131: Again, please use more appropriate verbal tenses

Line 122: TEM abbreviation should be defined

Lines 135-137: sentence could benefit from a better rearrangement of the elements, check English grammar

Line 138: the detail of N2 freezing is unnecessary an inaccurate…freezing of samples avoids DNA degradation (and other detrimental effects) but also helps with grinding and homogenization

Line 139Please correct “secquenced”

Line 142: “was used”, also Genome Analysis Toolkit software should be referenced

Line 143: AMFI don’t understand why “analysis” appears in italics

Line 146: MAF abbreviation should be defined

Line 146: ED abbreviation should be defined

Line 157: please correct “snalysis”

Line 160: please correct verbal tenses

Line 165 Please correct “plooen”

Line 168: Reverse-transcription methodology should be referenced or detailed

Line 170 Please correct “wasset”

Line 176: Please correct “plooen”

Line 179: the methodology used to obtain the sequencing library from the RNA samples should be referenced or described.

Line 180: trimmomatic and Hisat2 software should be referenced

Line 187: Please correct “plooens”

Line 192: “metabolites”

  • Regarding Results

Line 196: “Results”

Line 198: EMS abbreviation should be defined the first time that appears in the text, in line 51

Line 199: “ms” abbreviation should be defined the first time that appers in the text, in line 94

Line 204: “ms” should appear in italics

Lines 221-224: Figure 1 legend: the age of the organ samples shown in a,c,d  should be detailed

Line 234: Again, I don’t understand why” pollen grains” or “surface depression” are in italics. Also, I think the correct term in this case will be “pollen grains were surface-depressed”

Line 251: please correct the font format of “sequencing Depth”

Line 252: “were detected”

Line 253: “was chosen”

Line 255: “snp index”, “In the two BSA data”

Line 256: LD abbreviation should be defined

Line 259-261: the sentence regarding CYAC1 involvement in sexual reproduction in Arabidopsis should be back up with an appropriate reference

Line 276: please correct font formats in the sentence

Line 281: please correct the term “metabolisms”: pollen tube growth, biological regulation or pollen tube development are not metabolism processes…

Line 286, line 307: again, I must insist that metabolites are not expressed. Metabolites have different levels of abundance

Line 287 and 292-293: “categories of metabolites”

Line 290 “17 Metabolic processes”

Line 293: “might be related”

Figure 4: scatter and volcano plots should be showed as supplemental figures, since they are not very informative about the significance of the results obtained

  • Regarding Discussion

Lines 341-342: a gene do not cause anything (again, I must insist that encodes for a protein that performs a task in the cell). In this sense, the authors should modify the expression “the GMS genetic 341 network caused by VrCYCA1” to a more appropriate

Lines 344, 357: I suppose that these lines are titles of sections of the discussion, so they should be appearing in bold font

Line 358: “qRT-PCR”

Line 379: “analyzed” and “expression” appear in italics

Lines 380-381: “the TATA-box “appears in italics (an element of a promotor is not a gene)

Line 381: “of” appear in italics

Lines 387,390, 408: I am not sure why this sentences appear in form of  a numbered list and bold font. For example… if I assume that "1)" is a section of the discussion where “The mechanism of pollen abortion of genetic male sterile mutant” is going to be proposed of discussed it strikes to me that only a brief phrase appears and of course do not appear any proposed mechanism in the discussion…

Line 411: “in” “metabolic” “processes “appear in italics

Lines 429-431: the authors have not discussed why “The difference contents of those metabolites between CK and ms “are an” important reason for the sterility of mung bean caused by VrCYCA1 and six related genes”

Line 434: EMS mutagenesis is a chemical method, not by radiation!

Lines 439-441: categories of metabolites are hardly “genetic reasons” of a phenotype

  • Regarding Supplemental data:

Lines 464-471: “Supplemental data”

Table S2: please use a unified letter size and format.

Author Response

Respond to the second reviewer

Comment: Manuscript entitled “Identification and clarification of VrCYCA1, a key genic male sterility related gene in mungbean by multi-omics analysis” by Liu et al describes a comparative study between EMS mutagenized-lines of mungbean with male-sterility phenotype (ms lines), and wild-type ones (CK). Their phenotype was compared by several approximations (pollen viability, citologila analysis of flower and pollen morphology), showing pollen. Also, the authors show that male-sterility trait segregation of ms lines is compatible with a recessive mutation on a single nuclear gene. The mutated gene is subsequently identified through a QTL study comparing sets of male-sterile and fertile plants. The authors show that the ms lines present a point mutation on the promoter region of VrCYCA1, which prevents its expression. The effects of the lack of VrCYCA1 expression are further examined in comparative studies of transcriptomes and metabolomes of flowers of MS and CK lines, where sets of genes and metabolites with differential expression or levels were identified. The authors perform enrichment analyses of the functional categories and pathways associated with these genes and metabolites in order to identify possible mechanisms responsible for the male sterility phenotype of VrCYCA1 mutant plants. The results obtained are interesting and add new data about the regulatory mechanisms implicated in male sterility in mungbean. The experimental design and the proposed approaches used in this work seem both correct for the most part. However, in my opinion, some aspects of the manuscript need to be revised before considering this work suitable for publication. Some of them are enumerated in the additional comments below:

Major revision:

Question 1 One of the main aspects that weighs down this version of the manuscript is the grammar and use of English. I have pointed out some suggestions in my additional comments below, but I strongly recommend a more exhaustive revision by a native English speaker.

Answer: Thanks very much for your comments and suggestions and in the revision we have addressed this issue as can as possible in the revision. And the manuscript would carefully checked thorough language editing by a professional before resubmit.

Question 2 Transcriptome and metabolome data should be uploaded to appropriate public databases.

Answer: Thanks very much for your comments and suggestions and the transcriptome data will be uploaded to NCBI. When completed we would offer the SUB ID. The metabolome data were list in the supplementary data set 6 with the peak area of each metabolite.

Question 3 Figures 3,4,5: image resolution is very poor:

Answer: Thanks very much for your comments and suggestions and in the revision we have addressed this issue. All the figures have a PDF version, and the picture quality is very high, if need we can offer the PDF versions.

Question 4 The text is illegible in most cases the discussion of work in this version of the manuscript is often rather superficial and should be improved. The authors are too inclined to repeat the description of the results again, and sometimes the authors stop in this point, instead of integrating them into previous results from the literature that helps to arrive at some new interpretation regarding the observed results. In this sense, I must make the authors two recommendations: 

A: To to place their results in the context of a broader framework of the previous literature. 

Answer: Thanks very much for your comments and suggestions and in the revision we have addressed this issue.

B: the sets of genes and metabolites detected in the different comparisons are also discussed in a very superficial way. The authors should try to further discuss the biological implications of the proteins encoded for the different genes detected, as well as the metabolites and pathways.

Answer: Thanks very much for your comments and suggestions. The functional verification of the VrCYCA1 will be identified using functional complementarity experiment of Arabidopsismutants and than verification of the VrCYCA1 will be performed through overexpression of VrCYCA1 and CRISPR technology. At the same time, the VrCYCA1 gene networks will be analyzing the transgenic materials, This work will be done in our next work. The discussion is based on the phenotypic differences. As lack of transgenic materials the, discussion of the proteins encoded for the different genes detected, as well as the metabolites and pathways was conservative.

Question 7:Statistical methodology and threshold criteria to consider significant changes is absent for gene expression and metabolite analyses is absent in material and methods section

Answer: Thanks very much for your comments and we have fixed them and revised the related sentences in the revision (lines117-119 and lines216-217).

Question 8 Expression levels of the genes that form part of the interaction network identified in (figure 7c and yd) should be also confirmed by qRT-PCR. By the other side, some expression changes of some genes (TCP4, CTF7) should not be described as significant (their P> 0,05)

Answer: Thanks very much for your comments and suggestions. The VrCYCA1 gene networks will be analyzing in the transgenic materials, This work will is doing in our next work.

For all these reasons, I consider that the present manuscript is still far from enough, in my opinion, to consider its publication in this journal.

Answer: Thanks very much for your comments and suggestions.

Additional comments.

Line 10: GMS should be defined

Answer: Thanks very much for your suggestions. We have fixed the related sentences in the revision (lines 9).

Lines 14-16: Please consider rephrase and join the two sentences regarding expression of VrCYCA1 gene

Answer: Thanks very much for your suggestions. We have fixed the related sentences in the revision (lines 15-16).

Lines 17-18: metabolites are synthetized or degraded but not “expressed”

Answer: Thanks very much for your suggestions. We have fixed the related sentences in the revision (lines 17).

Lines 18-20: genes are segments of DNA that code for proteins, these proteins perform tasks. Hence the genes are not involved in anything of had interactions with other genes…. please rephrase accordingly

Answer: Thanks very much for your suggestions. We have fixed the related sentences in the revision (lines 18-19).

Lines 21-24 please check grammar of the sentence

Answer: Thanks very much for your suggestions. We have fixed the related sentences in the revision (lines 21-24).

Line 28: “Mining plant GMS-related genes “ I suggest that the authors make this modification when referring to GMS-relates genes as in lines 31, 33 and so on

Answer: Thanks very much for your suggestions. We have fixed the related sentences in the revision (lines 29, 33, 35, 37).

Lines 37, 39: again, these genes code for proteins involved in these processes…please modify the sentence accordingly

Answer: Thanks very much for your suggestions. We have fixed the related sentences in the revision (lines 37, 42).

Lines 41: “much less about…”!

Answer: Thanks very much for your suggestions. We have fixed the related sentences in the revision (lines 42).

Line 48 “GWAS is generally applicable”

Answer: Thanks very much for your suggestions. We have fixed the related sentences in the revision (lines 49).

Line 50: BC, DH, RIL or NAM abbreviations have not been defined

Answer: Thanks very much for your suggestions. We have fixed the related sentences in the revision (lines 51-54).

Lines 51-52: please check the use of verbal tenses in” To date, thousands of reported QTN/QTL involved in 51 quantitative traits, have been identified by GWAS and linkage analysis”.

Answer: Thanks very much for your carefulness. We have fixed the related sentences in the revision (lines 54-55).

Line 64: MER3 should appear in italics

Answer: Thanks very much for your carefulness. We have fixed the related sentences in the revision (lines 68).

Line 63: “…used this method to identify…”

Answer: Thanks very much for your carefulness. We have fixed the related sentences in the revision (lines 67).

Line 66-67: I don’t understand why “analysis trends” and large genetic interval” appear in italics

Answer: Thanks very much for your carefulness. We have fixed the related sentences in the revision (lines 69-70).

Line 68: in this case (Liu et al 2020a) should be removed since is already references with “[32]”

Answer: Thanks very much for your carefulness. We have fixed the related sentences in the revision (lines 72).

Lines 68-70 please include a brief description ot the work of liut el at 2020a to inform to the reader at least about which plant/process t in particular are studying these authors.

Answer: Thanks very much for your carefulness. We have fixed the related sentences in the revision (lines 72-75)

Lines 73-74: Again, “(Fiehn 202)” is already referenced by [35]

Answer: Thanks very much for your carefulness. We have fixed the related sentences in the revision (lines 80).

Lines 77-80: Please check grammar od the sentence

Answer: Thanks very much for your carefulness. We have fixed the related sentences in the revision (lines 83-85)

Lines 81 and 247: I am not very sure that “consists” should be used in these sentences

Answer: Thanks very much for your carefulness. We have fixed the related sentences in the revision (lines 87-94, and lines 268) 

Line 86-87: “were establishes by identification of differentially-expressed genes”

Answer: Thanks very much for your carefulness. We have fixed the related sentences in the revision (lines 94-95) 

Line 88: “According to the results obtained” 

Answer: Thanks very much for your carefulness. We have fixed the related sentences in the revision (lines 96) 

Line 94: Regarding Materials and methods the origin of “ms” lines should be detailed or referenced. Were these EMS-mutagenized lines generated in this work or have they already been described in a previous manuscript?

Answer: Thanks very much for your carefulness. We have fixed the related sentences in the revision (lines 102-106) 

Line 95: Please check “ofin”

Answer: Thanks very much for your carefulness. We have rewrote the sentences in the revision (lines 102-106) 

Lines 99-100: I think that the methodology of section 2.2 should be detailed more accurately what means “After removed the 99 phenotypic with non-segregated individuals,”

Answer: Thanks very much for your carefulness. We have rewrote the sentences in the revision (lines 108-119) 

Line: 106: Please use the appropriate verbal tenses: “flower buds and blooming flowers of “Weilyu11” and ms mutants were collected at different development stages during….”

Answer: Thanks very much for your carefulness. We have rewrote the sentences in the revision (lines 120-130) 

Line 114: DAF abbreviation should be defined

Answer: Thanks very much for your suggestions. We have rewrote the sentences in the revision (lines 132) 

Line 116: FAA abbreviation should be defined

Answer: Thanks very much for your suggestions. We have rewrote the sentences in the revision (lines 136) 

Line 124: PBS abbreviation should be defined

Answer: Thanks very much for your suggestions. We have rewrote the sentences in the revision (lines 145) 

Lines 126-127: please use mora appropriate verbal tenses

Answer: Thanks very much for your suggestions. We have rewrote the sentences in the revision (lines 142-144) 

Lines 130-131: Again, please use more appropriate verbal tenses

Answer: Thanks very much for your suggestions. We have rewrote the sentences in the revision (lines 148-150) 

Line 122: TEM abbreviation should be defined

Answer: Thanks very much for your suggestions. We have rewrote the sentences in the revision (lines 151-152) 

Lines 135-137: sentence could benefit from a better rearrangement of the elements, check English grammar

Answer: Thanks very much for your suggestions. We have rewrote the sentences in the revision (lines 154-159) 

Line 138: the detail of N2 freezing is unnecessary an inaccurate…freezing of samples avoids DNA degradation (and other detrimental effects) but also helps with grinding and homogenization

Answer: Thanks very much for your suggestions. We have rewrote the sentences in the revision (lines 160) 

Line 139Please correct “secquenced”

Answer: Thanks very much for your suggestions. We have rewrote the sentences in the revision (lines 160) 

Line 142: “was used”, also Genome Analysis Toolkit software should be referenced

Answer: Thanks very much for your suggestions. We have rewrote the sentences in the revision (lines 164) 

Line 143: AMFI don’t understand why “analysis” appears in italics

Answer: Thanks very much for your carefulness. We have rewrote the sentences in the revision (lines 165) 

Line 146: MAF abbreviation should be defined

Answer: Thanks very much for your carefulness. We have rewrote the sentences in the revision (lines 168) 

Line 146: ED abbreviation should be defined

Answer: Thanks very much for your carefulness. We have rewrote the sentences in the revision (lines 167) 

Line 157: please correct “snalysis”

Answer: Thanks very much for your carefulness. We have rewrote the sentences in the revision (lines 179-183) 

Line 160: please correct verbal tenses

Answer: Thanks very much for your carefulness. We have rewrote the sentences in the revision (lines 185-187) 

Line 165 Please correct “plooen”

Answer: Thanks very much for your carefulness. We have rewrote the sentences in the revision (lines 192) 

Line 168: Reverse-transcription methodology should be referenced or detailed

Answer: Thanks very much for your carefulness. We have rewrote the sentences in the revision (lines 195-203) 

Line 170 Please correct “wasset”

Answer: Thanks very much for your carefulness. We have rewrote the sentences in the revision (lines 198) 

Line 176: Please correct “plooen”

Answer: Thanks very much for your carefulness. We have rewrote the sentences in the revision (lines 205) 

Line 179: the methodology used to obtain the sequencing library from the RNA samples should be referenced or described.

Answer: Thanks very much for your carefulness. We have rewrote the sentences in the revision (lines 209-217) 

Line 180: trimmomatic and Hisat2 software should be referenced

Answer: Thanks very much for your carefulness. We have rewrote the sentences in the revision (lines 210 and ) 

Line 187: Please correct “plooens”

Answer: Thanks very much for your carefulness. We have rewrote the sentences in the revision (lines 220 ) 

Line 192: “metabolites”

Answer: Thanks very much for your carefulness. We have rewrote the sentences in the revision (lines 224-228 ) 

Line 196: “Results”

Answer: Thanks very much for your carefulness. We have rewrote the sentences in the revision (lines 232 ) 

Line 198: EMS abbreviation should be defined the first time that appears in the text, in line 51

Answer: Thanks very much for your carefulness. We have rewrote the sentences in the revision (lines 54) 

Line 199: “ms” abbreviation should be defined the first time that appers in the text, in line 94

Answer: Thanks very much for your carefulness. We have rewrote the sentences in the revision (lines 98) 

Line 204: “ms” should appear in italics

Answer: Thanks very much for your carefulness. We checked it, “ms” was in italics.

Lines 221-224: Figure 1 legend: the age of the organ samples shown in a,c,d  should be detailed

Answer: Thanks very much for your carefulness. We have rewrote the sentences in the revision (lines 256-259) 

Line 234: Again, I don’t understand why” pollen grains” or “surface depression” are in italics. Also, I think the correct term in this case will be “pollen grains were surface-depressed”

Answer: Thanks very much for your carefulness. We have rewrote the sentences in the revision (lines 271) 

Line 251: please correct the font format of “sequencing Depth”

Answer: Thanks very much for your carefulness. We have rewrote the sentences in the revision (lines 288) 

Line 252: “were detected”

Answer: Thanks very much for your carefulness. We have rewrote the sentences in the revision (lines 289) 

Line 253: “was chosen”

Answer: Thanks very much for your carefulness. We have rewrote the sentences in the revision (lines 289) 

Line 255: “snp index”, “In the two BSA data” 

Answer: Thanks very much for your carefulness. We have rewrote the sentences in the revision (lines 292) 

Line 256: LD abbreviation should be defined

Answer: Thanks very much for your carefulness. We have rewrote the sentences in the revision (lines 293) 

Line 259-261: the sentence regarding CYAC1 involvement in sexual reproduction in Arabidopsis should be back up with an appropriate reference

Answer: Thanks very much for your carefulness. We have rewrote the sentences in the revision (lines 298) 

Line 276: please correct font formats in the sentence

Answer: Thanks very much for your carefulness. We have rewrote the sentences in the revision (lines 298) 

Line 281: please correct the term “metabolisms”: pollen tube growth, biological regulation or pollen tube development are not metabolism processes…

Answer: Thanks very much for your carefulness. We have rewrote the sentences in the revision (lines 319) 

Line 286, line 307: again, I must insist that metabolites are not expressed. Metabolites have different levels of abundance

Answer: Thanks very much for your carefulness. We have rewrote the sentences in the revision (lines 314) 

Line 287 and 292-293: “categories of metabolites”

Answer: Thanks very much for your carefulness. We have rewrote the sentences in the revision (lines 331) 

Line 290 “17 Metabolic processes”

Answer: Thanks very much for your carefulness. We have rewrote the sentences in the revision (lines 328) 

Line 293: “might be related”

Answer: Thanks very much for your carefulness. We have rewrote the sentences in the revision (lines 328) 

Figure 4: scatter and volcano plots should be showed as supplemental figures, since they are not very informative about the significance of the results obtained

Answer: Thanks very much for your suggestions. Scatter and volcano plots presented a clearly fold-change and t-test statistic, we hope to show here

Lines 341-342: a gene do not cause anything (again, I must insist that encodes for a protein that performs a task in the cell). In this sense, the authors should modify the expression “the GMS genetic network caused by VrCYCA1” to a more appropriate

Answer: Thanks very much for your suggestions. We have rewrote the sentences in the revision (lines 371) 

Lines 344, 357: I suppose that these lines are titles of sections of the discussion, so they should be appearing in bold font

Answer: Thanks very much for your suggestions. We have rewrote the sentences in the revision (lines 386-390) 

Line 358: “qRT-PCR”

Answer: Thanks very much for your suggestions. We have rewrote the sentences in the revision (lines 406-312) 

Line 379: “analyzed” and “expression” appear in italics

Answer: Thanks very much for your suggestions. We have rewrote the sentences in the revision (lines 428-429) 

Lines 380-381: “the TATA-box “appears in italics (an element of a promotor is not a gene)

Answer: Thanks very much for your suggestions. We have rewrote the sentences in the revision (lines 430) 

Line 381: “of” appear in italics

Answer: Thanks very much for your suggestions. We have rewrote the sentences in the revision (lines 431) 

Lines 387,390, 408: I am not sure why this sentences appear in form of  a numbered list and bold font. For example… if I assume that "1)" is a section of the discussion where “The mechanism of pollen abortion of genetic male sterile mutant” is going to be proposed of discussed it strikes to me that only a brief phrase appears and of course do not appear any proposed mechanism in the discussion…

Answer: Thanks very much for your suggestions. We have rewrote the sentences in the revision (lines 406-412, lines 427-436, lines 442-445, lines 453-454, lines 463-466, lines 427-436, lines 471-472, lines 488-490, and lines 493-497.

Line 411: “in” “metabolic” “processes “appear in italics

Answer: Thanks very much for your suggestions. We have corrected the sentences in the revision (lines 471) 

Lines 429-431: the authors have not discussed why “The difference contents of those metabolites between CK and ms “are an” important reason for the sterility of mung bean caused by VrCYCA1 and six related genes”

Answer: Thanks very much for your suggestions. We have corrected the sentences in the revision (lines 488-490 and lines 493-498). And, the discussion is based on the phenotypic differences. As lack of transgenic materials the, discussion was conservative.

Line 434: EMS mutagenesis is a chemical method, not by radiation!

Answer: Thanks very much for your suggestions. We have corrected the sentences in the revision (lines 501) 

Lines 439-441: categories of metabolites are hardly “genetic reasons” of a phenotype

Answer: Thanks very much for your suggestions. We have corrected the sentences in the revision (lines 508) 

Lines 464-471: “Supplemental data”

Answer: Thanks very much for your suggestions. We have corrected the sentences in the revision (lines 531-538) 

Table S2: please use a unified letter size and format.

Answer: Thanks very much for your suggestions. We have corrected the table S2 in a suitable format.

Reviewer 3 Report

  1. The English language is to be improved, some of the corrections are made in the reviewed PDF copy
  2.  In the Methodology, the generation of mutants using EMS needs to be described
  3. Other corrections mentioned PDF copy attached 

Author Response

Respond to the third reviewer

Additional comments.

Question 1 Line 10: Expand when writing for the first time line 84.

Answer: Thanks very much for your comments and we have fixed them and revised the related sentences in the revision (lines 84).

Question 2 A brief description of EMS treatments and LD50 value need to be provided line 94.

Answer: Thanks very much for your comments and we have fixed them and revised the related sentences in the revision (lines 95-99).

Question 3 The fertile plants from the same plant as the sterile plants were harvested per plant line 98.

Answer: Thanks very much for your comments and we have fixed them and revised the related sentences in the revision (lines 101-104). 

Question 3 After removed the phenotypic with non segregated individuals line 99.

Answer: Thanks very much for your comments and we have fixed them and revised the related sentences in the revision (lines 105-106). 

Question 4 Collect should be collected in line 106

Answer: Thanks very much for your comments and we have revised the related sentences in the revision (lines 111). 

Question 5 ms should be write in italics in line 106.

Answer: Thanks very much for your comments and we have revised the related sentences in the revision (lines 111). 

Question 6 "To investigate the pollen detail morphology of the ms lines" should be "To investigate the detailed morphology of pollens of the ms" in lines 122.

Answer: Thanks very much for your comments and we have revised the related sentences in the revision (lines 127). 

Question 7. Transfer should be transferred in line 124.

Answer: Thanks very much for your comments and we have revised the related sentences in the revision (lines 129). 

Question 8. Treat should be treated in line 138.

Answer: Thanks very much for your comments and we have revised the related sentences in the revision (lines 144). 

Question 9. Writing is not clear. rephrase the sentences in line 154-156.

Answer: Thanks very much for your comments and we have revised the related sentences in the revision (lines 161-168). 

Question 10. Pollen should be pollens in line 165, 176 and 187.

Answer: Thanks very much for your comments and we have revised the related sentences in the revision (lines 146, 187 and 199). 

Question 11. Sentence is not clear/confusing rewrite in line 194-195.

Answer: Thanks very much for your comments and we have revised the related sentences in the revision (lines206-207). 

Question 12. Normally should be norma in line 205.

Answer: Thanks very much for your comments and we have revised the related sentences in the revision (lines 217). 

Question 13. Those should be these in line 216.

Answer: Thanks very much for your comments and we have revised the related sentences in the revision (lines 228). 

Question 13. Inconsistent font type and size in line 251.

Answer: Thanks very much for your comments and we have revised the related sentences in the revision (lines 264). 

Question 14. Detect should be detected in line 252.

Answer: Thanks very much for your comments and we have revised the related sentences in the revision (lines 265). 

Question 15. Choose should be chosen in line 253.

Answer: Thanks very much for your comments and we have revised the related sentences in the revision (lines 266). 

Question 16. Delete the words that are striked out here in line 263.

Answer: Thanks very much for your comments and we have revised the related sentences in the revision (lines 277). 

Question 17. Heterosis utilization in mung bean here not suitable in line 344.

Answer: Thanks very much for your comments and we have revised the related sentences in the revision (lines 362-366). 

Question 18. Significant reduction of VrCYCA1 expression led to GMS in mungbean here not suitable in line 357.

Answer: Thanks very much for your comments and we have revised the related sentences in the revision (lines 381-386, 394-397, 401-410 and 442-444). 

Question 19. Conduct should be conducted in line 374.

Answer: Thanks very much for your comments and we have revised the related sentences in the revision (lines 266). 

Reviewer 4 Report

  1. The English writing should be improved.
  2. “To date many genes have ……” . Add a comma after date.
  3. Line30-31. The description of these two sentences is repetitive and confusing. Rewrite it and merge into one sentence.
  4. There should be a space between “20” and “h”.
  5. There should be a space between “15” and “mins”. What’t the mins? Should be min or minutes.
  6. What’t the O? Zero or letter o?
  7. Line124 2h; line125 15min; line126 1-2h; line127 15min; line129 15min; line131 30s; line179 1μg; line256 20Kb. The problems are same as Number 4. Please double check!
  8. What’t the “4 ℃refrigerator”? Check.
  9. treat shoule be treated.
  10. “ms”. The word is italic in some places and regular in others, please double check.
  11. The first letter of after should be capitalized.
  12. Line171-173. “Step 1.......95℃”. The description of sentences is confusing. Rewrite.
  13. add “of” between mg and powder.
  14. “EMS (ethyl methane sulfonate)” should be “ethyl methane sulfonate (EMS)”
  15. add “lines” after 6.
  16. add * or ** in the fugure.
  17. What’s the “g enetic”? double check.
  18. add “lines” after 42.
  19. Add more detailed chart notes about Figure 2. What’s the a, b, c.....l?
  20. “P” should be lowercase and italic, please check other places.
  21. Line285-289. The description of sentences is confusing. Rewrite.
  22. The resolution of Figure 3,4,5,6 is not enough to see the font, which has a distinct gray background. Please provide better figures.
  23. The layout of figure 4 is not good, please do it again.
  24. Line344,357. is it subtitle? If yes, please bold.
  25. Consider identifying the function of VrCYCA1, transient expression or stable expression. Verify the effects of the gene mentioned in the article on six genes and some metabolites.

Author Response

Respond to the fourth reviewer

Comment:This study describes a series of approaches to identify the disrupted genes and mechanisms leading to male sterility in a mutagenised line of mung bean. The wider significance of this study is the potential applicability of this mutant as part of hybrid breeding. The start of the introduction gives an up-to-date account of recent findings in male sterility research in model plants and the state of knowledge in the focal species, mung bean. The introduction then gives useful background on the approach.

Major revision:

Question 1 The study aims need rewriting for clarity with new acronyms spelled out.

Answer: Thanks very much for your comments and we have fixed them and revised the related sentences in the revision (lines 11; lines 21-23; lines 96-97).

Question 2 The methods section on plant material is difficult to follow and needs rewriting.

Answer: Thanks very much for your comments and we have fixed them and revised the related sentences in the revision (lines100-105).

Question 3 Important information on DNA methods is missing.

Answer: Thanks very much for your comments. In the current version, we added the DNA methods in the revision (lines154-159).

Question 4 The description of BSA pools is confusing.

Answer: Thanks very much for your comments and we have fixed them and revised the related sentences in the revision (lines175-182).

Question 5 More RNAseq information is needed.

Answer: Thanks very much for your comments and we have fixed them and revised the related sentences in the revision (lines196-1204 and lines216-219 ).

Question 6 The metabolomic data analysis is not given.

Answer: Thanks very much for your comments and we have fixed them and revised the related sentences in the revision (lines225-229 and lines 232-234).

Question 7 Results of extra phenotypes are provided in the results that are not present in the methods. Either remove these sections or include these extra measures as an extra study aim.

Answer: Thanks very much for your comments and we have fixed them and revised the related sentences in the revision (lines115-118).

Question 8 The BSA results are difficult to interpret due to confusion about the pools being compared. The criterion for focusing on a stable QTL observed across multiple BSA comparisons should be added to the methods. 

Answer: Thanks very much for your comments and we have fixed them and revised the related sentences in the revision. BSA pools were redescribed in lines175-178. 

And stable QTL observed across multiple BSA comparisons were added to the method in lines180-183.

Question 9 Differential expression of RNAseq and metabolites both need to be described in the methods.

Answer: Thanks very much for your comments and we have fixed them and revised the related sentences in the revision (lines 217-219 ,and lines 232-234).

Question 10 The qPCR experiment of interacting genes was not described in the methods.

Answer: Thanks very much for your comments and we have fixed them and revised the related sentences in the revision (lines169-203).

Question 11 Large parts of the discussion simply repeat the results and these sections should be shortened and replaced with interpretation and reference to the literature.

Answer: Thanks very much for your comments and suggestions and in the revision we have addressed this issue. (lines 392-395, lines 409-415, lines 429-435, lines 437-439, lines 445-447, lines 455-457, lines 466-469, lines 473-474, lines 483-492 and lines 496-499)

Question 12 It is difficult to follow the English language in many places and considerable revision is required.

Answer: Thanks very much for your comments and suggestions and in the revision we have addressed this issue as can as possible in the revision. And the manuscript would carefully checked thorough language editing by a professional before resubmit.

Specific comments

L2 I'm not sure what you mean by "clarification" in the title. Perhaps replace with a more widely used term.

Answer: Thanks very much for your suggestions. "Identification and clarification of VrCYCA1, a key genic male sterility related gene in mungbean by multi-omics analysis", here "clarification" means "analysis". Maybe "analysis" is more suitable (lines 2).

L9-10 The link between fertility and heterosis is not clear from this statement. Explain GMS acronym at first use.

Answer: Thanks very much for your suggestions. We have fixed the related sentences in the revision (lines 9).

L12 use the singular term "locus" here (and elsewhere)

Answer: Thanks very much for your suggestions. We have fixed the related sentences in the revision (lines 12).

L16 Drop "And" and "which"

Answer: Thanks very much for your suggestions. We have fixed the related sentences in the revision (lines 15).

L17 Insert "differentially" between "significantly expressed"

Answer: Thanks very much for your suggestions. We have fixed the related sentences in the revision (lines 17).

L21-22 Rewrite this statement for clarity.

Answer: Thanks very much for your suggestions. We have fixed the related sentences in the revision (lines 21-22).

L23 Rather than focus on heterosis here, i suggest changing the focus to male sterility.

Answer: Thanks very much for your suggestions. We have fixed the related sentences in the revision (lines 23).

L27 Replace "trait" with "tool"

Answer: Thanks very much for your suggestions. We have fixed the related sentences in the revision (lines 27).

L40-41 Give full latin species name of mung bean at first use.

Answer: Thanks very much for your suggestions. We have fixed the related sentences in the revision (lines 41).

L50 Explain the acronyms that you introduce here.

Answer: Thanks very much for your suggestions. We have fixed the related sentences in the revision (lines 50-51).

L63 Insert "and" between "method identified"

Answer: Thanks very much for your suggestions. We have fixed the related sentences in the revision (lines 64).

L64-65 MutMap method needs to be explained here.

Answer: Thanks very much for your suggestions. We have fixed the related sentences in the revision (lines 68-69).

L69-70 Rewrite in present tense to reflect that you are discussing general methods here.

Answer: Thanks very much for your suggestions. We have fixed the related sentences in the revision (lines 68-69).

L79 Rather than a theoretical advance, I would say that this is more of a practical advance.

Answer: Thanks very much for your suggestions. We have fixed the related sentences in the revision (lines 83).

L81-84 Rewrite this important study aim for clarity and explain acronyms. Split into at least two sentences.

Answer: Thanks very much for your suggestions. We have fixed the related sentences in the revision (lines 85-88).

L84 Not clear what two results you are referring to. So far i only see a two sample BSA comparison.

Answer: Thanks very much for your suggestions. We have fixed the related sentences in the revision (lines 89-90).

L87 Replace "differential" with "differentially expressed"

Answer: Thanks very much for your suggestions. We have fixed the related sentences in the revision (lines 92).

L89 Why do you say "could reveal" here. Do you feel that evidence is still lacking?

Answer: Thanks very much for your suggestions. As lack of evidence form transgenic experiments, here we prudent to say "could reveal".

L93 Start the methods with some lines about the origin of both the cultivar and ms plant material.

Answer: Thanks very much for your suggestions. We have fixed the related sentences in the revision (lines 100-105).

L98-99 I don't understand rewrite for clarity. How were GMS individuals reproduced across generations?

Answer: Thanks very much for your suggestions. We have fixed the related sentences in the revision (lines 108-110).

L100-101 Replace "phenotypic" with "population"? Clarify if only a single segregating population was identified for further analysis. Multiple populations could have distinct independent mutations contributing to the GMS phenotype.

Answer: The fertile plants from the same plant as the sterile plants were harvested per plant, the heterozygous fertile plants in M2 were used to breeding M3, and the M4 were breeding form M3 by the same methods and planted according to the pedigree. And the heterozygous fertile plants in M2 were a single segregating population, the M3 and M4 were breeding from the single segregating of a heterozygous fertile plants in M2. Though, we also used more than one BSA populations to find the positive site(lines 108-110).

L106-107 Rewrite in past tense. Be consistent with cultivar name; either Weilyu1 or CK. Specify the flowering stages sampled, with references as necessary. This is important for repeatability.

Answer: Thanks very much for your suggestions. We have fixed the related sentences in the revision (lines 124, lines 126-129).

L121-133 It seems like section 2.5 is a continuation of section 2.4. Merge these sections as "electron microscopy analysis". Rewrite instructions as past tense throughout. Write out TEM acronym.

Answer: Thanks very much for your suggestions. We have fixed the related sentences in the revision (lines 130; lines 138-150 ).

L138 The methods is missing any detail on DNA extraction or DNA library construction.

Answer: Thanks very much for your suggestions. We have fixed the related sentences in the revision (lines 153-158 ).

L145-148 Write out ED and MAF acronyms. How can MAF and missing rate be used as part of BSA? How many libraries were being compared? State the pools that were compared with CK.

  • Write out ED and MAF acronyms

Answer: Thanks very much for your suggestions. We have fixed the related sentences in the revision (lines 164-165).

  • How can MAF and missing rate be used as part of BSA?SNP

Answer: MAF is the Minor Allele Frequency. It can be used to exclude SNPs which are not informative because they show little variation in the sample set being analyzed. For instance, if a SNP shows variation in only 1 of the 89 individuals, it is not useful statistically and should be removed.

  • How many libraries were being compared?

Answer: Short reads secquenced by Illumina HiSeq 4000 platform (with an average sequencing depth ≥50X. Individuals >30 mixed DNA were sequenced. One male  pool and one fertile plants pool both compared with CK.

Fekih, R.; Takagi, H.; Tamiru, M.; Abe, A.; Natsume, S.; Yaegashi, H.; Sharma, S.; Kanzaki, H.; Matsumura, H.; Saitoh, H.; et al. MutMap+: genetic mapping and mutant identification without crossing in rice. PLoS One 2013, 10, e68529.

Wenger, J.W.; Schwartz, K.; Sherlock, G. Bulk segregant analysis by high-throughput sequencing reveals a novel xylose utilization gene from Saccharomyces cerevisiae. PLoS Genet 2010, 6, e1000942.

L149-150 State the size of the sliding window. 

Answer: Thanks very much for your suggestions. We have fixed the related sentences in the revision (lines 169-170). ΔSNP index was calculated by sliding window (20Kb, 20Kb, linkage disequilibrium value was at about 20Kb).

L153-156 Move the description of pools to the start of this section. I am confused by the pool description. Each BAS pool consists of three types of plants that include a mix of GMS and fertile plants. I understand BSA to be a paired pool comparison. Rewrite for clarity.

Answer: Thanks very much for your suggestions. We have fixed the related sentences in the revision (lines 175-181). There were two BSA pools, one consisted of the CK (Weilyu11) pool (1), the male sterile plants in M3 pool (2) and the fertile plants in M4 pool (4) and other consisted of the CK (Weilyu11) pool (1), the male sterile plants in M4 pool (3) and the fertile plants in M4 pool (4). The fertile plants in M4 pool (4) were used as the fertile plants pools for the two experiments. Significant QTNs were common obtained by both of the two BSA pools snalysis results. And stable QTN observed across multiple BSA comparisons, singificant QTNs both identified in the (1), (2) (4); and (1), (3) (4) pools were consider as the reliable QTNs. Fertile and male pools first compared to the CK, then did the paired comparison.

L156-157 Rewrite for clarity.

Answer: Thanks very much for your suggestions. We have fixed the related sentences in the revision (lines 175-182).

L160 Replace "significantly" with "significant". Name the reference genome that was used.

Answer: Thanks very much for your suggestions. We have fixed the related sentences in the revision (lines 180; lines 182). 

L160-163. What tools were used for annotation.

Answer: Thanks very much for your suggestions. We identified the genes or their Arabidopsis homologous genes, used Arabidopsis genes annotation.

L163-170 This section is missing description of primer design and testing. 

Answer: Thanks very much for your suggestions. We add the primers information in the table S2 (196-203).

L163-165 Replace "significantly" with "differential" Explain DEG acronym L165 Typo "plooen" and elsewhere. Is it "pollen"?

Answer: Thanks very much for your suggestions. We have fixed the related sentences in the revision (lines 186; lines 191). Yes, typo "plooen" and elsewhere. Is "pollen".

L168-169 Name the product used for reverse transcription.

Answer: Thanks very much for your suggestions. We have fixed the related sentences in the revision (lines 198). 

L170 Typo "wasset" to "was set"

Answer: Thanks very much for your suggestions. We have fixed the related sentences in the revision (lines 198). 

L173-174 How many replicate samples were measured?

Answer: Thanks very much for your suggestions. We have fixed the related sentences in the revision (lines 202). 

L178-180 Name the kit or reference the method used to build the RNA sequencing libraries. How many replicate libraries were produced and sequenced?

Answer: Thanks very much for your suggestions. We have fixed the related sentences in the revision (lines 206-211). 

L180-181 What cleaning criteria were used?

Answer: Thanks very much for your suggestions. We used Hisat2 to did the blast. Programs were download form https://daehwankimlab.github.io/hisat2/ (lines 2153).

D Kim, Langmead B , Salzberg S L. HISAT: A fast spliced aligner with low memory requirements[J]. Nature Methods, 2015, 12(4). And we added the reference in the new submission.

L182 Why was the reference genome rather than a reference transcriptome used for mapping?

Answer: Thanks very much for your suggestions. As, reference transcriptome was not stable in different analysis. We usually used the reference genome for blasting.

L183-184 How were RPKM (spell out acronym) measures analysed to determine differential expression? Were only candidate genes tested? 

Answer: Thanks very much for your suggestions. RPKM formula:

And, all genes were analysed. We have fixed the related sentences in the revision (lines 215-219). 

L185 Typo "Metabonomic" to "Metabalomic" 

Answer: Thanks very much for your suggestions. We have fixed the related sentences in the revision (lines 220). 

L186-187 How many replicates were prepared and analysed?

Answer: Thanks very much for your suggestions. Three pollen samples of CK and ms were taken for detection. We have fixed the related sentences in the revision (lines 233-2340). 

L191-192 What classes of metabolites were targeted? You need to mention how the data was analysed subsequently. How were these results linked to the DNA and RNA data? L198-200 Which generation are you referring to here; M2

Answer: Thanks very much for your suggestions. Both classes of metabolites in the Madison Metabolomics Consortium Database were analyzed.

As the metabolites analysis, the peak areas of the metabolites were blast with MMCD. And we added the reference in the new submission.

The plooen of ms (M4 pool) and CK (M4 pool) at 0 DAF, which were used in the RNA-seq.. We have fixed the related sentences in the revision (lines 232-234).  

L201-202 You need to add phenotypic analysis to the methods.

Answer: Thanks very much for your suggestions. We have fixed the related sentences in the revision (lines 126-129). 

L210 I don't understand the term "round abortion".

Answer: Thanks very much for your comments.

Round abortion refers to the round appearance of pollen grains without stained starch grains.

L211 Typo "g enetic" to "genetic", "separation" to "segregation" (also in following lines). Table 1 can be dropped as the important values are summarized in the text. Figure 1 Legends for panels a and c are swapped. No need to include scale bar lengths as already shown in the figure panels. 

Answer: Thanks very much for your suggestions. We have fixed the related sentences in the revision (lines 126-129). Table 1 scale, bar lengths we reserved as the incomplete expression in the text. We have fixed the related sentences about figure 1 Legends.

L226 Paraffin method is not included in the methods. Figure 2 legend. Typo "cotyledon" You are showing anthers here. Explain the different developmental stages that you are showing in panels a - f.

Answer: Thanks very much for your suggestions. Paraffin method were mentioned in line 131-137. Figure 2a 2c, 2e both used to describe pollen, maybe it small in Figure 2a. A-f were the flowers at 0 DAF. We have fixed the related sentences in the revision (lines 287).

L246-248 I think it needs to be explained, probably in the methods, that the male fertile lines are heterozygous for the ms allele unless you only chose individuals that did not show ms segregation in their progeny. This could affect your delta SNP and ED analysis interpretations as the expected difference in frequency between pools is less.

Answer: Thanks very much for your suggestions. We have fixed the related sentences in the revision (lines 108-118). The ms plants individuals were homozygous, for significant phenotype.

L251-253 It should be the SNPs relative to the reference genome rather than relative to CK/Weilyu1 that are presented as these are of interest as potential causal mutations.

Answer: Thanks very much for your suggestions. We checked in carefully, the content, ‟And, 500315 and 53529 SNPs (Supplemental Date S1-2) were detect in the M3 and M4 BSA data, respectively, compared to Sulv 1 genome, which was choose as the reference genome.”was ok.

L254-255 What are the two BSA data? The criterion for focusing on a stable QTL observed across multiple comparisons should be added to the methods.

Answer: Thanks very much for your suggestions. We checked in carefully, related sentences in the revision (lines 175-183).

L255-256 Reference the LD statistic. Explain LD acronym. 

Answer: Thanks very much for your suggestions. LD statistic were form our analysis: 196 mungbean accessions (Supplementary Data Set 1),with an average depth of ~15.1×. After removing SNPs with an average coverage depth<8× and with minor allele frequency (MAF) less than 5%, we identified 3607508 SNPs, 2110609 short indels (≤40 bp). The average SNP density was 7.4 SNP/Kb (Figure 1a), and a high-quality sequence of Sulv 1 (elite cultivar) about 473.67Mb was selected as the reference genome (N50 = 42.35 Mb, and 95.7% genes could be functionally annotated). Then, we performed the linkage disequilibrium (LD) analysis, and found that the LD dropped to half of its maximum value at 20.1 Kb, 11.3Kb and 14.3Kb for the resequenced population in landrace, wild and all (landrace + wild) accessions, respectively. However, the paper hasn't accepted. Related research was none.

LD acronym were in lines 198-199.

L258 What test is this p value summarizing?

Answer: Thanks very much for your suggestions. We checked in carefully, p value here was a mistake. We have fixed the related sentences in the revision (lines 298). 

L260-261 Reference this report.

Answer: Thanks very much for your carefulness. We have fixed the related sentences in the revision (lines 301). 

L261-266 These expression results should be moved to the next results section. 

Answer: Thanks very much for your carefulness. We have fixed the related sentences in the revision (lines 322-327). 

L261-263 Figure 3d suggests that other genes also had differential RNAseq expression. Include the p value results of all genes in the figure.

Answer: Thanks very much for your carefulness. We have fixed the related figure in the revision. 

L263-266 I would also add the direction of the difference of expression when summarizing these results.

Answer: Thanks very much for your carefulness. We have fixed the related sentences in the revision (lines 326). 

L280-281 Drop "And". Replace "metabolisms" with "metabolic functions". Functional annotation approach should be described in the methods.

Answer: Thanks very much for your carefulness. We have fixed the related sentences in the revision (lines 329). Functional annotation approach were in lines 332-333.

L284 Typo "Date" to "Data".

Answer: Thanks very much for your carefulness. We have fixed the related sentences in the revision (lines 330) and other place.

L286 Differential metabolite analysis needs to be described in the methods.

Answer: Thanks very much for your carefulness. We have fixed the related sentences in the revision (lines229-231). MMCD database include metabolite categorization and pathway enrichment information (http://mmcd.nmrfam.wisc.edu/). And we added the reference in the new submission.

L287 Metabolite categorization/pathway enrichment analysis needs to be described in the methods. Figure 4 labels are very small. Enlarge if possible.

Answer: Thanks very much for your carefulness. We have fixed the related figure in the revision. 

L299-300 No vertical lines shown in panel 4b. Description of panel c matches panel d and missing description of panel c. Figure 5 labels are very small.

Answer: Thanks very much for your carefulness. We have fixed the related figure in the revision. All the figures have a PDF version, and the picture quality is very high, if need we can offer the PDF versions.

L310-312 This part of the legend is repeated from the legend for figure 5. One of them is incorrect.

Answer: Thanks very much for your carefulness. L310-312 should be 165.

Figure 6. Pathway enrichment analysis corresponding to the differential metabolites. Statistical results of 165 significantly differential concentration metabolites.

L317-319 The qPCR experiment of interacting genes needs to be described in the methods.

Answer: Thanks very much for your comments. We have fixed the related sentences in the revision.Here, we used the transcriptome data. And, the functional verification of the VrCYCA1 will be identified using functional complementarity experiment of Arabidopsismutants and than verification of the VrCYCA1 will be performed through overexpression of VrCYCA1 and CRISPR technology. At the same time, the VrCYCA1 gene networks will be analyzing the transgenic materials, This work will be done in our next work. The discussion is based on the phenotypic differences.

L322 I don't understand what is meant by "medium confidence value" here.

Answer: Thanks very much for your carefulness. The statistic for PPI were described on STRING (https://cn.string-db.org/), and 0.4 was medium confidence value.

L342 Replace "caused by" to "that includes"

Answer: Thanks very much for your carefulness. We have fixed the related sentences in the revision (lines 372). 

L344 Change this heading to refer to GMS, which is what you actually studied. 

Answer: Thanks very much for your carefulness. We have fixed the related sentences in the revision (lines 391).

L346-347 Replace "abortion" with "aborted".

Answer: Thanks very much for your carefulness. We have fixed the related sentences in the revision (lines 398).

L351-353 This example is poorly linked to the current study. Are you suggesting combining these traits as part of a mung bean breeding programme?

Answer: Thanks very much for your carefulness. We have fixed the related sentences in the revision (lines 409).

L357 You can be more specific than "change" here. Use "reduction" instead.

Answer: Thanks very much for your carefulness. We have fixed the related sentences in the revision (lines 410).

L358-378 This whole section only repeats the results. Link what you have found to the related literature on this topic.

Answer: Thanks very much for your carefulness. We have rewrote the sentences in the revision (lines 411-415, and lines 429-436).

L367-368 I don't understand what you mean here about compliance.

Answer: Thanks very much for your carefulness. We have rewrote the sentences in the revision (lines 422-424).

L369 Typo "though" to "through"

Answer: Thanks very much for your carefulness. We have fixed the related sentences in the revision (lines 424).

L379 Typo "Than" to "Then"

Answer: Thanks very much for your carefulness. We have fixed the related sentences in the revision (lines 437-439).

L387, 390-391, 408-409 Inconsistent use of numbered sections 

Answer: Thanks very much for your carefulness. We have fixed the related sentences in the revision (lines 391, 449, and 471). 

L387-389 Merge this section with the next or delete it since it is only one line long. 

Answer: Thanks very much for your carefulness. We have fixed the related sentences in the revision (lines 466-469).

L393-397 Remove repeated lists of gene functions.

Answer: Thanks very much for your carefulness. We have fixed the related sentences in the revision (lines 457-46).

L412-414 This sentence is missing important text after "metabolites".

Answer: Thanks very much for your carefulness. We have fixed the related sentences in the revision (lines 445-447).

L422-429 This list of metabolite level differences belongs in the methods.

Answer: Thanks very much for your carefulness. We have fixed the related sentences in the revision (lines 232-234).

L429-431 Beyond observing these differences, you do not have evidence that they cause GMS. Restate this conclusion that they are associated with your ms phenotype.

Answer: Thanks very much for your carefulness. We have fixed the related sentences in the revision (lines 467-469).

L435 Typo "separation" to "segregation"

Answer: Thanks very much for your carefulness. We have fixed the related sentences in the revision (lines 503).

Author Response

(The authors gave the same response as above.)
